# Towards classification of 5d SCFTs: Single gauge node

Patrick Jefferson[1,2], Hee-Cheol Kim[1,3], Cumrun Vafa[1] and Gabi Zafrir[4]

**1** Jefferson Physical Laboratory, Harvard University, Cambridge, MA 02138, USA
**2** Center for Theoretical Physics, Department of Physics, Massachusetts Institute of Technology, Cambridge, MA 02139
**3** Department of Physics, POSTECH, Pohang 37673, Korea
**4** Kavli IPMU (WPI), UTIAS, the University of Tokyo, Kashiwa, Chiba 277-8583, Japan

## Abstract

We propose a number of apparently equivalent criteria necessary for the consistency of a 5d SCFT in its Coulomb phase and use these criteria to classify 5d SCFTs arising from a gauge theory with simple gauge group. These criteria include the convergence of the 5-sphere partition function; the positivity of particle masses and monopole string tensions; and the positive definiteness of the metric in some region in the Coulomb branch. We find that for large rank classical groups simple classes of SCFTs emerge where the bounds on the matter content and the Chern-Simons level grow linearly with rank. For classical groups of rank less than or equal to 8, our classification leads to additional cases which do not fit in the large rank analysis. We also classify the allowed matter content for all exceptional groups.

# 1   Introduction

One of the main achievements of string theory has been a deeper understanding of quantum field theories. We have learned, in particular, of the existence of non-trivial superconformal field theories, not only in dimensions $d \leq 4$ but also in $d = 5, 6$. Studying the cases with $d > 4$ is interesting for a number of reasons: These theories turn out to involve, not only interactions of massless particles, but also of tensionless strings, which are quite novel. Higher dimensional SCFTs also seem to be simpler and in particular classifiable—in particular, the $(2, 0)$ SCFTs in $d = 6$ are believed to be classified by ADE [1] and a classification for the $d = 6$, $(1, 0)$ SCFTs has been proposed [2–4]. Another motivation for studying cases with $d > 4$ is that higher dimensional theories can lead to lower dimensional ones by compactifications and flowing to the IR. This relationship between lower dimensional theories and their higher dimensional 'parents' also leads to the emergence of dualities in lower dimensional theories as a manifestation of the different possible ways of assembling the intermediate manifold from elementary pieces.

It is thus natural to ask if we can also classify the $d = 5$ SCFTs. String theory suggests that this class of theories can be obtained geometrically as M-theory on non-compact Calabi-Yau 3-folds ($CY_3$) where some compact 4-cycles have shrunk to a point. Unfortunately at present these singularities are not fully classified, although some progress has been made in this regard [5]. Rather than attempting to classify these theories in terms of singularities, we instead take a clue from the classification of 6d (1,0) theories, for which various geometric conditions were ultimately distilled into constraints on gauge theoretic data. It was found that the allowed 6d (1,0) theories can be viewed as generalized quiver theories. In that case, the classification was split into two parts: first, a classification of the quiver node types (i.e. the single node case), followed by a specification of the rules according to which the quiver nodes can be connected. There are a few exceptional cases where the quiver nodes had no gauge theory associated with them, which were nevertheless reachable by Higgs branch deformations of some gauge theory. One may hope that a similar story can play out in the case of 5d SCFTs. An example of this structure is the class of $SU(2)$ gauge theories coupled to matter studied in [6–10], where it was found that $SU(2)$ gauge theories with up to 7 fundamental

hypermultiplets can arise in a SCFT and are related to del Pezzo surfaces shrinking inside a $CY_3$. The case of $\mathbb{P}^2$ is the only case which is not realizable directly in terms of gauge theory (formally, this theory could be interpreted as '$SU(2)$ with -1 fundamental hypermultiplets'). But even this case can be obtained from gauge theory by starting from $SU(2)$ without matter and theta angle $\theta = \pi$, and passing through a strong coupling point where some massless modes emerge, are given mass, and integrated out [7, 11]. In a sense, this is similar to the exceptional cases in the 6d (1,0) classification.

Motivated by this analogy, we thus take seriously the idea of using gauge theoretic constructions as a window into the classification all 5d SCFTs. This approach was pioneered by Intriligator, Morrison, and Seiberg (IMS) in [8]. The idea in that work is that a weakly-coupled gauge system in its Coulomb phase can emerge in the IR as a deformation of an SCFT, so a classification of allowed IR theories would translate to a classification of UV fixed points, which correspond (in gauge theoretic terms) to the regime where the gauge coupling is infinite and all Coulomb branch parameters are set to zero. The assumption in [8] was that such an emergent IR gauge theory should be physically sensible for all values of the Coulomb branch parameters. More precisely, the condition that the vector multiplet scalars have a positive definite kinetic term on the entire Coulomb branch was imposed. This condition is equivalent to requiring that the matrix of effective couplings on the Coulomb branch is positive semi-definite. Using this criterion, quiver gauge theories were ruled out and theories with a simple gauge group coupled to matter were fully classified. However, it was found using geometric constructions [10] as well as brane constructions in Type IIB [12] that not only are quiver gauge theories allowed, but also that these quiver theories are sometimes dual to gauge theories with a simple gauge group. The loophole in the positivity condition of [8] ends up being the assumption that the gauge theoretic description should be valid for the entire Coulomb branch, since it could nevertheless be possible that for some values of the Coulomb branch parameters massless nonperturbative states emerge that introduce boundaries where the effective description breaks down. In some cases, these boundaries can be crossed at the expense of introducing a different description of the physics. Indeed, by using brane constructions and instanton analysis, new examples of gauge theories incompatible with the IMS criterion were found in [13–17], even in cases involving simple gauge group.

We propose a modification of the IMS criterion that only imposes the condition of positive definiteness of the metric to a subregion of Coulomb branch, rather than the entire Coulomb branch. However, it turns out that there are additional conditions that need to be checked: for example, 5d $\mathcal{N} = 1$ gauge theories theories have monopole strings, and the regions on the Coulomb branch where these strings become tensionless should be regarded as parts of a boundary where the effective gauge theory description breaks down. This is to say there may exist regions for which the metric is positive definite but some string tensions are nevertheless negative, which indicates that those regions are not physically sensible from the standpoint of a given effective description.

Therefore, the most natural criteria necessary for the consistency of allowed gauge theories are positive semi-definiteness of the metric on some subregion of the Coulomb branch, and furthermore positivity of masses and tensions of all physical degrees of freedom in that region. Even though this seems to be the most natural set of criteria, we find that there are three apparently simpler criteria, each of which we conjecture is equivalent to the previous statement: 1) the metric is positive-definite somewhere on the Coulomb branch, even if the string tensions are negative; 2) the tensions are positive in some region irrespective of metric positivity, as it seems that positivity of the metric in the same region automatically holds; and 3) the perturbative prepotential is positive on the *entire* Coulomb branch. The motivation for this last criterion is the convergence of the five-sphere partition function, which requires positivity of the perturbative prepotential. However, the surprise is that to get an equivalent

condition to the other criteria, we need to check this positivity of the prepotential even in the excluded regions where the naive tensions or metric eigenvalues may be negative. Why all these criteria are equivalent to the existence of a region of Coulomb branch admitting positive definite metric and positive mass (and tension) degrees of freedom is unclear, but we have found no counterexamples to this assertion in the numerous examples we have explored.

The organization of this paper is as follows. In Section 2, we review some relevant facts concerning 5d $\mathcal{N} = 1$ supersymmetric gauge theories. In particular, we explain in detail our assertion that the IMS criterion is incomplete. In Section 3, we propose our new (necessary) criteria for the existence of 5d SCFTs. We present our criteria in the form of a series of conjectures we believe to be equivalent. In Section 4, we classify non-trivial SCFTs with simple gauge group using our conjectures. We record the complete classification in a collection of several tables indicating the gauge group, matter content, classical Chern-Simons level (if applicable), and global symmetry groups. In Section 5, we discuss the connection between 6d SCFTs compactified on a circle and a class of 5d gauge theories we have identified. Finally, in Section 6, we conclude with a summary of our results and prospects for future research associated to this project. We summarize our conventions relevant to representation theoretic aspects of this paper in Appendix A; in Appendix B, we explain our rationale for excluding certain matter representations from our classification; and in Appendix C we discuss the five-sphere partition function along with its precise relation to the perturbative prepotential.

## 2  Effective Prepotentials on the Coulomb branch

We begin by presenting salient features of supersymmetric gauge theories in five-dimensions. The miminal SUSY algebra in 5d is the $\mathcal{N} = 1$ algebra, which contains eight real supersymmetries. A large class of 5d $\mathcal{N} = 1$ gauge theories can be engineered by $(p, q)$ five-brane webs in Type IIB string theory [12, 18, 19] and also by M-theory compactified on Calabi-Yau threefolds [7–10].

A 5d gauge theory with a gauge group $G$ has a Coulomb branch of moduli space where the real scalar field $\phi$ in the vector multiplet $\Phi$ takes nonzero vacuum expectation values in the Cartan subalgebra of the gauge group $G$. On a generic point of the Coulomb branch, therefore, the gauge symmetry is broken to its maximal torus $U(1)^{r_G}$ where $r_G = \mathrm{rank}(G)$, and therefore the Coulomb branch $\mathcal{C}$ is isomorphic to a fundamental Weyl chamber, $\mathcal{C} \cong \mathbb{R}^{r_G}/W(G)$, where $W(G)$ is the Weyl group of $G$. The low-energy abelian theory is governed by a prepotential $\mathcal{F}(\phi_i)$, which is a cubic polynomial in $\phi_i$. The perturbative quantum correction to the prepotential is exactly determined by an explicit one-loop computation [20]. With matter hypermultiplets in generic representations $\mathbf{R}_f$, the exact prepotential of the low-energy abelian theory is given by [6, 8]

$$\mathcal{F}(\phi) = \frac{1}{2g_0^2} h_{ij} \phi_i \phi^j + \frac{\kappa}{6} d_{ijk} \phi_i \phi^j \phi^k + \frac{1}{12} \left( \sum_{e \in \mathrm{root}} |e \cdot \phi|^3 - \sum_f \sum_{w \in \mathbf{R}_f} |w \cdot \phi + m_f|^3 \right), \quad (1)$$

where $g_0$ is the classical gauge coupling, $h_{ij} = \mathrm{Tr}(T_i T_j)$, $m_f$ are the hypermultiplet masses, $\kappa$ is the classical Chern-Simons level, and $d_{ijk} = \frac{1}{2}\mathrm{Tr}_{\mathbf{F}}(T_i\{T_j, T_k\})$, where $\mathbf{F}$ denotes the fundamental representation. Note that $d_{ijk}$ is non-zero only for $G = SU(N \geq 3)$. Gauge invariance requires the classical Chern-Simons level to be quantized as $\kappa + \frac{1}{2} \sum_f c_{\mathbf{R}_f}^{(3)} \in \mathbb{Z}$ where $c_{\mathbf{R}_f}^{(3)}$ is the cubic Dynkin index of the representation $\mathbf{R}_f$ defined by the relation $\mathrm{Tr}_{\mathbf{R}_f}(T_i\{T_j, T_k\}) = 2c_{\mathbf{R}_f}^{(3)} d_{ijk}$. The last two terms in parentheses are one-loop contributions coming from, respectively, the massive vector multiplets and massive hypermultiplets on the Coulomb branch. Due to the

absolute values in the prepotential, the Coulomb branch is divided into distinct sub-chambers, and the prepotential takes different expression in each sub-chamber.

The effective gauge coupling is given by a second derivative of the prepotential, and the metric on the Coulomb branch is set by the effecitve gauge coupling:

$$(\tau_{\text{eff}})_{ij} = \left(g_{\text{eff}}^{-2}\right)_{ij} = \partial_i \partial_j \mathcal{F}, \quad ds^2 = (\tau_{\text{eff}})_{ij} \, d\phi_i \, d\phi^j. \tag{2}$$

The BPS spectrum for 5d $\mathcal{N} = 1$ theories includes gauge instantons, electric particles and magnetic monopole strings. The central charges of the electric particles and monopole strings are, respectively

$$Z_e = \sum_{i=1}^{r_G} n_e^{(i)} \phi_i + m_0 I, \quad Z_m = \sum_{i=1}^{r_G} n_m^{(i)} \phi_{Di}, \tag{3}$$

where $n_e^{(i)}, n_m^{(i)} \in \mathbb{Z}$, $I$ is the instanton number, and the dual coordinates $\phi_{Di}$ are defined by way of the prepotential:

$$\phi_{Di} = \partial_i \mathcal{F}(\phi). \tag{4}$$

Bearing in mind that not every choice of integer coefficients $n_e^{(i)}, n_m^{(j)}$ in (3) corresponds to the central charge of a physical BPS state, we note that for any physical choice of coefficients $n_e^{(i)}, n_m^{(j)}$ the corresponding particle masses and monopole string tensions should be positive. Geometrically, because the tension of a monopole string is proportional to the volume of a 4-cycle, we can expect that there are at least as many independent monopole strings as there are 4-cycles. The number of independent monopole strings is therefore bounded below by $r_G$. At low energies, we expect that the number of independent strings is equal to the number of independent normalizable Kähler classes, implying that there are precisely $r_G$ independent monopole tensions. We note that it is always possible to choose a coordinate basis for which the independent physical string tensions are

$$T_i(\phi) \equiv \partial_i \mathcal{F}(\phi) = \phi_{Di}, \quad i = 1, \ldots, r_G. \tag{5}$$

In this paper, we will use the Dynkin basis (i.e. the basis of fundamental weights), in which we expect the tensions of $r_G$ independent monopole strings are captured by the above expression. The rationale for this choice is that a classical string tension in the Dynkin basis is a root associated to gauge algebra and can thus be taken to have positive projection on the Weyl chamber. (In the gauge theory description, these correspond to monopole solutions.) We therefore expect that the quantum-corrected string tensions are also positive in this basis.

It was argued in [8] that when the quantum part of the metric is non-negative on entire Coulomb branch, a gauge theory can have non-trivial renormalization group fixed point at UV. Otherwise, the kinetic term may become negative at some point on the Coulomb branch, thereby preventing the existence of a UV fixed point. This necessary condition for the existence of 5d interacting quantum field theories is equivalent to the condition that $\mathcal{F}(\phi)$ be convex on all of $\mathcal{C}$ and, viewed in this fashion, imposes interesting constraints on the list of possible gauge theories. For example, quiver gauge theories are ruled out, since for such theories there is always a region on $\mathcal{C}$ for which the metric is negative. Thus, it was concluded in [8] that only gauge theories with simple gauge groups can have non-trivial UV fixed points. The Coulomb branch metrics of gauge theories with simple gauge groups were studied in relation to this condition and a full classfication was given in [8]. All the theories in this classification are expected to have interacting fixed points in either 5d or 6d.

However, string dualities relate some theories predicted by the criterion in [8] to theories (such as quiver gauge theories) excluded by the same criterion. For example, an $SU(3)$ gauge

theory with $\kappa = 0$ and $N_{\mathrm{F}} = 2$ is dual to an $SU(2) \times SU(2)$ gauge theory, as explained in [12]. The metric of the former theory is positive definite on the entire Coulomb branch, while the latter theory has negative metric somewhere on the Coulomb branch. We discuss the details of this example below.

In fact, the existence of many interesting 5d field theories beyond the IMS bounds was argued by studying the dynamics of branes in string theory [13, 14, 17, 21, 22], $CY_3$-compactifications in M-theory [23–25], and also instanton analysis in 5d gauge theories [15, 16, 26]. For example, the authors in [8] predict that an $SU(N)$ gauge theory can couple to $N_{\mathrm{F}}$ fundamental hypermultiplets within the bound $N_{\mathrm{F}} \leq 2(N - |\kappa|)$. However, recently, the existence of the interacting fixed points for $SU(N)$ gauge theories with $N_{\mathrm{F}} \leq 2(N - |\kappa|) + 4$ was argued in various literature [13–16]. These theories are obviously beyond the bound in [8]. See also [13–17, 21–26] for many other examples.

This tells us that the condition of the metric being positive semi-definite on the entire Coulomb branch is too strong, and therefore we may be able to relax the resulting constraints on the matter content and find a new classification which involves all known consistent 5d field theories.

Indeed, we find that the previous constraint on the metric on the Coulomb branch is too strong. This was already noticed for the simplest $SU(2) \times SU(2)$ quiver gauge theory, for instance, in [12]. The $SU(2) \times SU(2)$ gauge theory has real two-dimensional Coulomb branch of the moduli space. The scalar fields $\phi_1, \phi_2$ in the respective vector multiplets of each gauge group parametrize the Coulomb branch of the moduli space. In the chamber with $\phi_1 > \phi_2 > 0$, the quantum prepotential of this theory is given by

$$\mathcal{F}(\phi) = \frac{1}{2g_1^2} \phi_1^2 + \frac{1}{2g_2^2} \phi_2^2 + \frac{1}{6} \left( 8\phi_1^3 + 8\phi_2^3 - (\phi_1 + \phi_2)^3 - (\phi_1 - \phi_2)^3 \right), \qquad (6)$$

where $g_1, g_2$ are the respective gauge couplings for the two $SU(2)$ gauge groups. Here we have set the mass parameters of the bi-fundamental hypermultiplet to zero for convenience.

Let us examine if the metric is negative somewhere in the Weyl chamber. The matrix of effective couplings is

$$\tau_{\mathrm{eff}}(\phi) = \begin{pmatrix} 1/g_1^2 + 6\phi_1 & -2\phi_2 \\ -2\phi_2 & 1/g_2^2 - 2\phi_1 + 8\phi_2 \end{pmatrix}. \qquad (7)$$

One can easily check that one of the eigenvalues of the metric becomes negative near $\phi_2 \sim 0.255\phi_1$ when the inverse couplings are small, $1/g_{1,2}^2 \ll 1$. Naively, this quiver theory cannot have an interacting fixed point according to the criterion of [8]. However, it turns out that this theory has a nice brane engineering in Type IIB string theory [12] as well as a geometric construction via compactification of M-theory on a $CY_3$ [10]. The five-brane web in Figure 1 will give rise to the 5d $SU(2) \times SU(2)$ gauge theory at low-energy.

It was pointed out in [12] that this theory has a singularity in the Coulomb branch owing to the appearance of massless instanton particles and tensionless monopole strings. Thus the $SU(2) \times SU(2)$ gauge theory description breaks down beyond the singularity. This singularity can be detected by computing the monopole string tensions explicitly. The tensions of the monopole strings are given by

$$T_1 = \partial_1 \mathcal{F} = \phi_1/g_1^2 + 3\phi_1^2 - \phi_2^2, \quad T_2 = \partial_2 \mathcal{F} = \phi_2/g_2^2 - 2\phi_1\phi_2 + 4\phi_2^2. \qquad (8)$$

In the brane configuration in Figure 1 monopole strings correspond to D3-branes wrapping two compact faces. Their tensions are proportional to the areas of two faces. We find that $T_1$ and $T_2$ are the same as the areas of the right and the left compact face respectively. As $\phi_2$ decreases, the monopole string coming from the D3-brane wrapping the left square of $\phi_2$

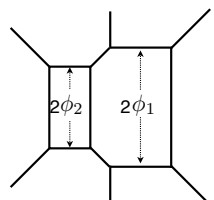

Figure 1: Five-brane web for the $SU(2) \times SU(2)$ gauge theory.

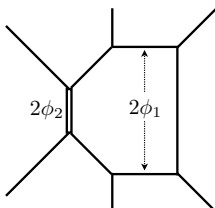

Figure 2: Appearance of massless degrees of freedom in the $SU(2) \times SU(2)$ gauge theory.

tends to be lighter and eventually becomes massless near $\phi_2 \sim \phi_1/2$ with $1/g_{1,2}^2 \ll 1$. It is clear that this point on the Coulomb branch where we meet the new light degrees of freedom is far away from the region where we encounter the negative metric. The prepotential in (6) cannot describe the dynamics of the system beyond the singularity due to the new light objects. The interior of the physical Coulomb branch of the $SU(2) \times SU(2)$ gauge theory is therefore specified by the bounds $\phi_1 > \phi_2 > \phi_1/2$. Notice, in particular, that this region is narrower than the naive Coulomb branch $\phi_1 > \phi_2 > 0$. The $SU(2) \times SU(2)$ gauge theory is an effective description of the system in Figure 1 only within the sub-region $\phi_1 > \phi_2 > \phi_1/2$.

If we want to study the precise physics around $\phi_2 = \phi_1/2$ we should use another effective description which has the light particles in its perturbative spectrum. In this case, the relevant description is the $SU(3)$ gauge theory with $N_F = 2$. The duality between the $SU(2) \times SU(2)$ gauge theory and the $SU(3)$ gauge theory was studied in [12, 27]. Therefore, based on this duality, since the $SU(3)$ gauge theory is a non-tivial quantum theory with 5d CFT fixed point, which is predicted already in [8], we can also expect that the $SU(2) \times SU(2)$ gauge theory is a non-trivial quantum theory. These two gauge theories are good low-energy effective descriptions in different parameter regimes of the same theory.

In this example, by exploiting the S-duality of Type IIB, we can clearly understand what happens when some non-perturbative degrees of freedom becomes massless. While the $SU(2) \times SU(2)$ gauge theory description breaks down near the point $\phi_2 = \phi_1/2$ due to the emergence of light instanton particles, the perturbative description of the $SU(3)$ gauge theory is reliable at the point. The light instantons at $\phi_2 = \phi_1/2$ give rise to non-abelian $SU(2)$ gauge symmetry enhancement in the dual $SU(3)$ gauge theory. It follows that the Coulomb branch of the $SU(3)$ gauge theory beyond the point $\phi_2 = \phi_1/2$ is merely a Weyl copy of the enhanced $SU(2)$ gauge symmetry. This means that the correct prepotential on the chamber $\phi_2 < 2\phi_1$ after the transition is a copy of the original prepotential on the Weyl chamber $\phi_2 > 2\phi_1$ obtained by exchanging $\phi_1 \leftrightarrow \phi_2 - \phi_1$. Therefore we expect a discontinuity of the prepotential in the $SU(2) \times SU(2)$ gauge theory at the transition point $\phi_2 = \phi_1/2$ detectable by a non-perturbative quantum correction to the prepotential despite the fact that the perturbative prepotential cannot capture this. With the prepotential corrected by non-perturbative effects, the metric of the

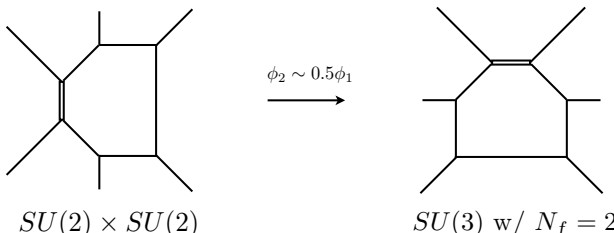

Figure 3: Duality of $SU(2) \times SU(2)$ theory and $SU(3)$ theory with two fundamental hypermultiplets, $N_{\mathbf{F}} = 2$.

$SU(2) \times SU(2)$ theory is always positive definite on the entire (physical) Coulomb branch.

The lesson from this simple example is that in general, the physical Coulomb branch may be smaller than the naive Coulomb branch identified in the perturbative description. Stated differently, the actual physical Coulomb branch cannot necessarily be identified with a fundamental Weyl chamber.

## 3 New Criteria

The above example shows that the IMS criterion is too restrictive. In view of the above discussion, we will relax the condition that the metric be positive everywhere on the perturbative Coulomb branch, since this condition imposes metric positivity even in unphysical regions. Our reasoning is as follows: in some regions of the perturbative Coulomb branch, there may exist boundaries where the non-perturbative degrees of freedom become massless or tensionless. Beyond these boundaries, these masses and tensions appear to become negative, which signals that the perturbative computation can no longer be trusted. We therefore restrict our attention to the physical regions in the Coulomb branch where all degrees of freedom have positive masses or tensions.

### 3.1 Main conjecture

We claim that if the metric is positive definite on the physical Coulomb branch the gauge theory has non-trivial interacting fixed point. For these theories we can reach their fixed points by taking the infinite coupling limit $g_0^2 \to \infty$. It is therefore crucial to determine the exact physical Coulomb branch that we will investigate. We claim that the physical Coulomb branch is the subset $\mathcal{C}_{\mathrm{phys}} \subseteq \mathcal{C}$ bounded by the set of hyperplanes where some non-perturbative degrees of freedom become massless and/or tensionless:

$$\mathcal{C}_{\mathrm{phys}} = \{\phi \in \mathcal{C} \,|\, T(\phi) > 0, m_I^2(\phi) > 0\}\,. \tag{9}$$

Here $m_I^2(\phi) > 0, T(\phi) > 0$ indicates that the masses and tensions of all non-perturbative degrees of freedom are positive at the point $\phi$. Furthermore, the metric on the above physical Coulomb branch must be positive-definite. Namely,

$$\tau_{\mathrm{eff}}(\phi) > 0\,, \quad \phi \in \mathcal{C}_{\mathrm{phys}}\,. \tag{10}$$

This condition should be true for all physically-sensible 5d theories, including quiver theories.

It is crucial to our classification that we are able to identify the hyperplanes where massless degrees of freedom arise on the naive Coulomb branch. However, from a purely field-theoretic standpoint, the task of identifying such hyperplanes is somewhat subtle because it requires a non-perturbative analysis to identify exact instanton spectrum. On the other hand, from a

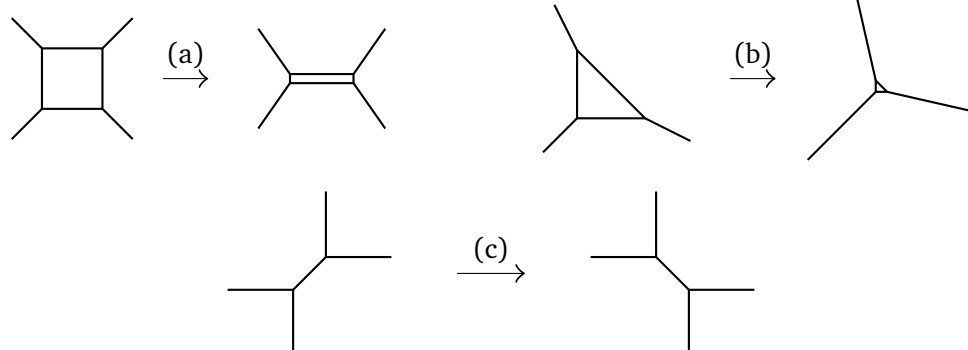

Figure 4: We illustrate the three types of geometric transitions occurring in Calabi-Yau threefolds using examples of $(p, q)$ five brane webs. Transition (a) corresponds to a 4-cycle shrinking to a 2-cycle, (b) corresponds to a 4-cycle shrinking to a point, and (c) is a flop transition obtained by first shrinking a 2-cycle to a point and blowing up another 2-cycle.

geometric perspective, it is in principle a straightforward exercise to identify the hyperplanes bounding the physical Coulomb branch $\mathcal{C}_{phys}$—from a geometric point of view, these are regions where we encounter a geometric transition in the $CY_3$. Therefore, we embark on a short digression to explain how to characterize these hyperplanes, and moreover their relation to the BPS spectrum.

We remark that there are three possible geometric transitions[1] associated to particles or monopole strings becoming massless: (a) a 4-cycle shrinking to a 2-cycle, (b) a 4-cycle shrinking to a point, and (c) a 2-cycle shrinking to a point. Examples of these three types of transitions in the case of $(p, q)$ five brane webs are depicted in Figure 4. We now describe these three types of transitions in more detail, and explain their relevance to our classification.

The first type of transition results in a local ADE singularity, as was argued in [28], so that there is always a gauge symmetry enhancement due to the appearance of a collection of massless W-bosons arising from the shrinking 2-cycle. The low energy dynamics in the vicinity of the singularity can therefore be captured precisely by a gauge theory description involving a simply-laced gauge group. The region beyond this wall should be regarded as an image generated by the Weyl group associated to the enhanced nonabelian gauge group, and is therefore unphysical.

The second type of transition should be distinguished from the first in the sense that a 4-cycle shrinking to a point always results in the appearance of an infinite number of massless particles associated to 2-cycles inside the shrinking 4-cycle. In this case, the low energy dynamics is described by a CFT with tensionless monopole strings interacting with massless electric particles [20]. The classic example of such a phenomenon is the local $\mathbb{P}^2$ shrinking to a point. From a geometric point a view, continuing past this kind of wall is prohibited and therefore corresponds to an unphysical region of the moduli space.

The third type of transition corresponds to a 2-cycle shrinking to a point, while all 4-cycle volumes remain finite. This type of transition is typically described as a flop transition in a geometric setting. When the volume of the 2-cycle corresponds to the mass of some perturbative BPS particle, this transition can be captured by the one-loop prepotential (1) and

---

[1]Note that 5d $\mathcal{N} = 1$ gauge theories can be constructed in string theory by compactifying M-theory on a $CY_3$ in which M2-branes wrap compact 2-cycles and M5-branes wrap compact 4-cycles in the $CY_3$. This setup leads to a 5d description in which BPS particle masses are controlled by the volumes of the 2-cycles, while monopole string tensions are controlled by the volumes of the 4-cycles.

manifests itself as a discontinuity of the cubic coefficients. However, when the volume of the 2-cycle corresponds to the mass of an instanton particle, the one-loop prepotential may not be able to detect this sort of transition at any point on the physical Coulomb branch. This sort of transition can in principle be captured by a careful analysis of the non-perturbative physics, such as a computation of the BPS partition function. Moreover, if a geometric description of the Calabi-Yau geometry is known, one can detect the presence of such a transition by computing the prepotential in terms of geometric data and carefully studying how the volumes of various 2-cycles change in the moduli space. An example of this is the geometric transition $\mathbb{F}_1 \to \mathbb{P}_2$ [7, 11].

The relationship between the volumes of 2-cycles (4-cycles) and the central charges $Z_e$ ($Z_m$) described in (3) implies that some combination of central charges vanishes for each transition type discussed above. Moreover, restrictions on the "allowed" linear combinations of 2- and 4-cycle classes in the geometric setting translate into charge quantization conditions in the field theory setting that ensure the corresponding BPS states all lie within physical charge lattices. Such constraints, for example, imply that the vanishing of irrational linear combinations of $\phi_i$ (i.e. combinations $\sum n_e^{(i)} \phi_i$ where $n_e^{(i)} \in \mathbb{R} \backslash \mathbb{Q}$) cannot correspond to geometric transitions. We will thus need to be careful to distinguish between physical and unphysical boundary hyperplanes in our analysis.

In this paper, we conjecture that by turning off all of the mass parameters,[2] we can identify $\mathcal{C}_{\text{phys}}$ by studying, in addition to flops by W-bosons, the first two types of transitions described above—namely those for which a 4-cycle shrinks. We expect that the one-loop prepotential can detect these types of transitions because the volumes of the string tensions are captured by partial derivatives with respect to the scalars $\phi_i$. The above conjecture also implies that the third type of transition can arise only as a pertubative flop transition by some massless electric particle when the mass parameters are turned off. Therefore, we are led to the main statement of this paper, concerning the characterization of what we refer to as the 'physical' Coulomb branch:

---

**Main Conjecture**

The *physical Coulomb branch* $\mathcal{C}_{\text{phys}}$ is defined by

$$\mathcal{C}_{\text{phys}} = \mathcal{C}_{T \geq 0}, \tag{11}$$

where

$$\mathcal{C}_{T \geq 0} = \{\phi \in \mathcal{C} \mid T(\phi) \geq 0\} \quad \text{and} \quad \partial \mathcal{C}_{\text{phys}} \text{ is } \textit{rational}. \tag{12}$$

Moreover, non-trivial 5d gauge theories have positive definite metric on the physical Coulomb branch:

$$\tau_{\text{eff}}(\phi) > 0, \quad \phi \in \mathcal{C}_{\text{phys}}. \tag{13}$$

---

Note that $\tau_{\text{eff}} > 0$ in the above equation indicates that $\tau_{\text{eff}}$ is a positive definite matrix, and *rational* means that the boundary $\partial \mathcal{C}_{\text{phys}}$ is specified by the setting some rational linear combinations of the Coulomb branch parameters equal to zero, i.e. $\sum_i n_i \phi_i = 0$, $n_i \in \mathbb{Z}$. In other words, we require that the boundary $\partial \mathcal{C}_{\text{phys}}$ is identified by the appearance of physical massless BPS particles, as discussed above. (Note that we find only five exceptional theories

---

[2]Note that 'mass parameters' refers collectively to both hypermultiplet mass parameters $m_f$ and inverse gauge couplings $m_0 = 1/g_0^2$, which are treated on the same footing from the 5d perspective—they are mass parameters for the global symmetries of the 5d theory.

where our classification becomes subtle due to the rationality condition. These five exceptional theories will be discussed in Section 4.4.) For the remainder of the paper, we use $\mathcal{C}_{\mathrm{phys}}$ to denote the subset of the Coulomb branch defined in (11), while we use $\mathcal{C}$ to denote the (naive) Coulomb branch isomorphic to a fundamental Weyl chamber. We will also use $\mathcal{C}_{T>0}$ to denote the interior of the physical Coulomb branch $\mathcal{C}_{\mathrm{phys}}$.

These new criteria can be used to classify all candidate non-trivial 5d QFTs with simple gauge groups. We emphasize that our main conjecture is a *necessary* (though perhaps not *sufficient*) set of criteria for the existence of a 5d interacting fixed point in the UV. Although brane configurations and geometric constructions can provide evidence confirming the existence of these theories, we will leave explicit realizations of the theories in our classification to future research. For the reason, it is important to bear in mind that our classification describes a maximal set of theories that can possibly exist, and the set of theories that actually exist may be a subset. For the remainder of this paper, we will be sloppy about the distinction between theories that satisfy the necessary set of conditions described by our conjectures and theories that have explicit realizations in string theory. In particular, we will refer to all candidates as 'non-trivial theories'.

We will classify these candidate non-trivial gauge theories below using our main conjecture for lower rank cases $r_G \leq 5$. For higher rank theories, our main conjecture appears to be rather inconvenient for the full classification of 5d QFTs. In the next sub-section, we will make three additional conjectures, each of which we believe is equivalent to the main conjecture described above, which make the problem of classification significantly more tractable. These additional conjectures impose constraints (involving at most one derivative of the prepotential), which are easier to implement in practice and thus enable us to make a full classification of arbitrary rank gauge theories with simple gauge groups. To our knowledge, our classification covers all known gauge theories with non-trivial fixed points and furthermore predicts large classes of new non-trivial 5d gauge theories that may exist. However, we would like to emphasize that only the main conjecture above is physically well-motivated at present. We do not have physical or mathematical arguments for the three additional conjectures described in the next sub-section.

Let us first demonstrate how to identify non-trivial QFTs and classify them using the Main Conjecture with simple examples. In this paper, we shall focus on the $\mathcal{N} = 1$ gauge theories without adjoint hypermultiplets.[3] We will also turn off all mass parameters except the gauge coupling.

We begin with $SU(3)$ gauge theories. An $SU(3)$ gauge theory has a two-dimensional Coulomb branch parametrized by scalar fields $\phi_1, \phi_2$. We find that an $SU(3)$ gauge theory can have hypermultiplets only in the fundamental or the symmetric representation. The dimensions of other $SU(3)$ representations are so large that the prepotential becomes negative at large gauge coupling when such hypermultiplets are included, and hence those representations must be excluded.

For some purposes, we will find it convenient to use Dynkin basis for roots and weights, while in some other cases it will prove more convenient to use the orthogonal basis. In the Dynkin basis, W-boson masses are given by the inner products $\phi \cdot \alpha_i$ with simple roots $\{\alpha_i\} = \{(2,-1),(-1,2),(1,1)\}$. In the Weyl subchamber $\mathcal{C}^{(1)} = \{\phi_1 > \phi_2 > 0\}$, we find the

---

[3]Note that if we add a single adjoint hypermultiplet to a 5d theory with a simple gauge group, the theory will be promoted to a $\mathcal{N} = 2$ gauge theory. All such 5d $\mathcal{N} = 2$ theories can be obtained from 6d $\mathcal{N} = (2,0)$ SCFTs on a circle $S^1$ with or without outer automorphism twists [29]. Gauge theories with two or more adjoint hypermultiplets have negative prepotentials, so they are not renormalizable.

prepotential of $SU(3)_\kappa + N_\mathbf{F}\mathbf{F} + N_\mathbf{S}\mathbf{S}$ is given by

$$\mathcal{F}_{SU(3)} = \frac{1}{g_0^2}(\phi_1^2 - \phi_1\phi_2 + \phi_2^2) + \frac{\kappa}{2}\left(\phi_1^2\phi_2 - \phi_1\phi_2^2\right) + \frac{1}{6}\left(8\phi_1^3 - 3\phi_1\phi_2^2 - 3\phi_1^2\phi_2 + 8\phi_2^3\right)$$
$$- \frac{N_\mathbf{F} - 9N_\mathbf{S}}{12}\left(2\phi_1^3 - 3\phi_1^2\phi_2 + 3\phi_1\phi_2^2\right). \tag{14}$$

The prepotential in the second Weyl chamber $\mathcal{C}^{(2)} = \{\phi_2 > \phi_1 > 0\}$ can be obtained by interchanging $\phi_1$ and $\phi_2$. The physical Coulomb branch is restricted to the sub-domain given by

$$\mathcal{C}_{\text{phys}}^{(1)} = \{\phi \in \mathcal{C}^{(1)} \,|\, T(\phi) > 0\}, \quad \mathcal{C}_{\text{phys}}^{(2)} = \{\phi \in \mathcal{C}^{(2)} \,|\, T(\phi) > 0\}, \tag{15}$$

in the first and the second Weyl subchamber, respectively. The condition for the gauge theory being a non-trivial theory is that the metric on $\mathcal{C}_{\text{phys}}^{(1)}$ and $\mathcal{C}_{\text{phys}}^{(2)}$ is positive in the infinite coupling limit $g_0^2 \to \infty$.

First consider the cases of $\kappa = 0, N_\mathbf{S} = 0$. Since $\kappa$ is an integer, we require $N_\mathbf{F}$ to be even. The theories with $N_\mathbf{F} \leq 6$ have a positive metric everywhere on the Coulomb branch, as discussed in [8], thus they are all non-trivial 5d QFTs. Therefore, we focus our attention on the cases with $N_\mathbf{F} > 6$. The eigenvalues of the metric and string tensions are drawn in Figure 5 for $N_\mathbf{F} = 6, 8, 10$. Unlike the case $N_\mathbf{F} = 6$ for which the metric always has positive eigenvalues, theories with $N_\mathbf{F} = 8, 10$ have a negative eigenvalue of the metric in certain sub-region of their Coulomb branches. However, one can easily check that their metrics are positive in the physical Coulomb branches defined by $T_1, T_2 > 0$. In other words, we encounter a singularity at $T_2 = 0$ in the naive Coulomb branch and the effective gauge theory description breaks down before we reach at the point where the metric becomes negative. Within the physical Coulomb branch $\mathcal{C}_{\text{phys}}$, the $SU(3)$ gauge theories with $N_\mathbf{F} = 8, 10$ have positive definite metrics. Thus we claim that they are good theories with non-trivial UV fixed points. They are good effective descriptions for the corresponding non-trivial theories at the fixed points in the parameter space $\mathcal{C}_{\text{phys}}$. We check that if we have $N_\mathbf{F} > 10$, the theories have no physical Coulomb branch, i.e. $\mathcal{C}_{\text{phys}} = \emptyset$, and thus these theories cannot have interacting fixed points. Indeed, it was conjectured in [14, 15] that the $SU(3)$ gauge theories with $N_\mathbf{F} \leq 10$ have non-trivial UV fixed points while the theories with $N_\mathbf{F} > 10$ may be inconsistent. Our computation provides concrete physical evidence for this conjecture.

Let us now turn on the Chern-Simons term at level $\kappa$. $SU(3)_{-1} + 8\mathbf{F}$ can be engineered in string theory by a $(p, q)$ 5-brane web [14]. Figure 6 is the 5-brane web diagram for this theory at zero masses in the first Weyl chamber $\phi_1 > \phi_2 > 0$. The two 5-branes at the top of the diagram end on 7-branes. Since the length of the first D5-brane is given by $1/g_0^2 - 2\phi_1$, this brane picture is a good description for the theory only within the parameter regime $1/g_0^2 \geq 2\phi_1$. When $1/g_0^2 < 2\phi_1$, two 7-branes at the top of the diagram need to be pulled inside the 5-brane loop.

At small coupling $1/g_0^2 \gg \phi_1$, the brane description is reliable. The monopole strings correspond to D3-branes wrapping the compact faces and the tensions of these strings $T_1, T_2$ are given by areas of two compact faces. It is straightforward to read off the tensions from the diagram and the results are

$$T_1 = \frac{1}{g_0^2}(2\phi_1 - \phi_2) + 2\phi_2(\phi_1 - \phi_2), \quad T_2 = \left(\frac{1}{g_0^2} + 2\phi_2 - \phi_1\right)(2\phi_2 - \phi_1). \tag{16}$$

One can easily check that these string tensions agree with the tensions obtained from the first derivatives of the prepotential $\mathcal{F}_{SU(3)}$ in (14). Although the brane description breaks down at large coupling beyond $1/g_0^2 = 2\phi_1$, the gauge theory description is still good. Thus we can expect that the above tension formula for the monopole strings may give the correct tensions of

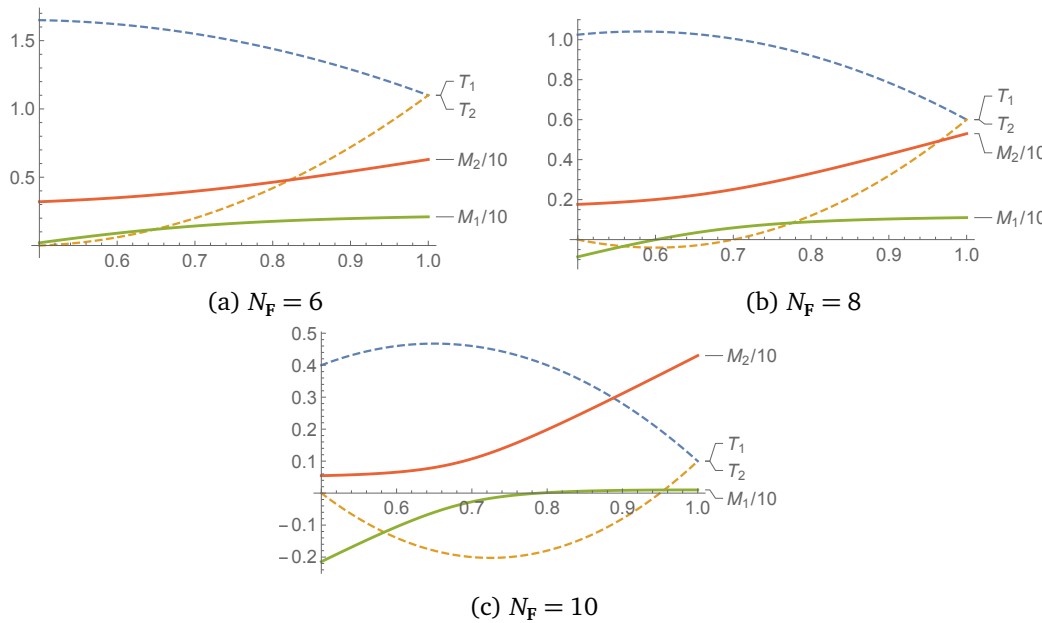

Figure 5: Solid lines are eigenvalues $M_1/10$ and $M_2/10$ of the metric and dashed lines are two string tensions $T_1$ and $T_2$ at $g^{-2} = 0.1$ for $N_{\mathbf{F}} = 6, 8, 10$. The horizontal axis is the ratio $\phi_2/\phi_1$. The physical Coulomb branches are $\mathcal{C}_{\text{phys}}^{N_{\mathbf{F}}=6} = \phi_1 \geq \phi_2 \geq \phi_1/2$, $\mathcal{C}_{\text{phys}}^{N_{\mathbf{F}}=8} = \phi_1 \geq \phi_2 \geq 0.7\phi_1$, and $\mathcal{C}_{\text{phys}}^{N_{\mathbf{F}}=10} = \phi_1 \geq \phi_2 \geq 0.95\phi_1$, respectively, with $\phi_1 \geq 0$.

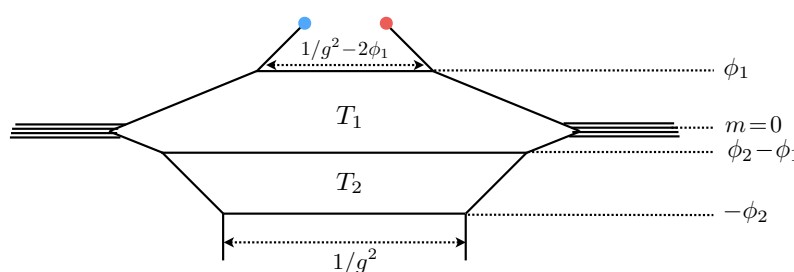

Figure 6: 5-brane web diagram for the theory $SU(3)_{-1} + 8N_{\mathbf{F}}$ at zero masses.

wrapped D3-branes even after 7-branes are moved inside the 5-brane loop at large coupling. In Figure 6, the bottom face denoted by $T_2$ shrinks to zero area when the last two internal D5-branes are placed on top of each other, i.e. $2\phi_2 - \phi_1 = 0$. Then the D3-brane wrapping the bottom face contributes tensionless strings to the 5d gauge theory. The string tensions are non-negative on the entire Coulomb branch in the first chamber $\phi_1 > \phi_2 > \phi_1/2$ even in the infinite coupling limit $g_0^2 \to \infty$. The non-negativity of the string tensions thus implies that the $SU(3)$ gauge theory and the 5-brane web are both good descriptions for this theory in the first chamber. This is consistent with the fact that the $SU(3)$ gauge theory has a positive metric on the first chamber as depicted in Figure 7a. In the second chamber, the metric becomes negative around $\phi_1 < 0.74\phi_2$. However, this is fine because the physical Coulomb branch is restricted to $\phi_2 > \phi_1 > 0.95\phi_2$ due to the tensionless strings. Therefore, as expected, $SU(3)_{-1} + 8\mathbf{F}$ is a non-trivial 5d QFT according to our new criteria.

Another interesting example is $SU(3)_{-4} + 6\mathbf{F}$. This is a new non-trivial 5d theory we predict using our new criteria.[4] This theory has the physical Coulomb branch only in the first chamber

---

[4]This theory is a marginal theory whose fixed point theory is not a 5d SCFT. We discuss marginal theories in

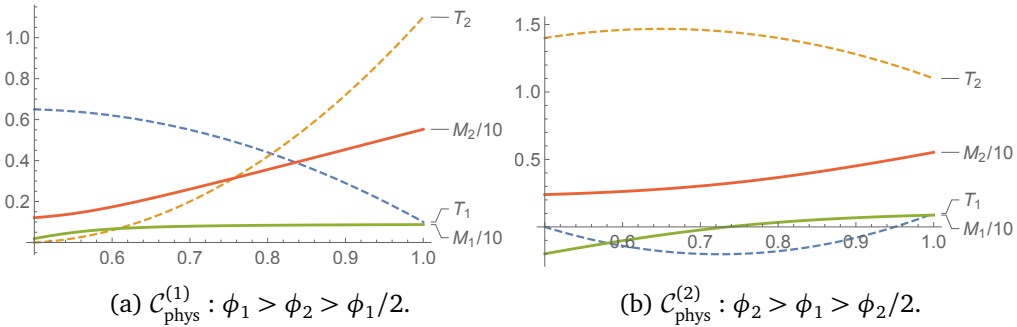

(a) $\mathcal{C}_{\text{phys}}^{(1)} : \phi_1 > \phi_2 > \phi_1/2$.      (b) $\mathcal{C}_{\text{phys}}^{(2)} : \phi_2 > \phi_1 > \phi_2/2$.

Figure 7: Eigenvalues of the metric (solid line) and string tensions (dashed lines) of $SU(3)_{-1} + 8\mathbf{F}$ fundamentals at $1/g_0^2 = 0.1$. The horizontal axes are ratios of scalar fields $\phi_2/\phi_1$ and $\phi_1/\phi_2$ respectively.

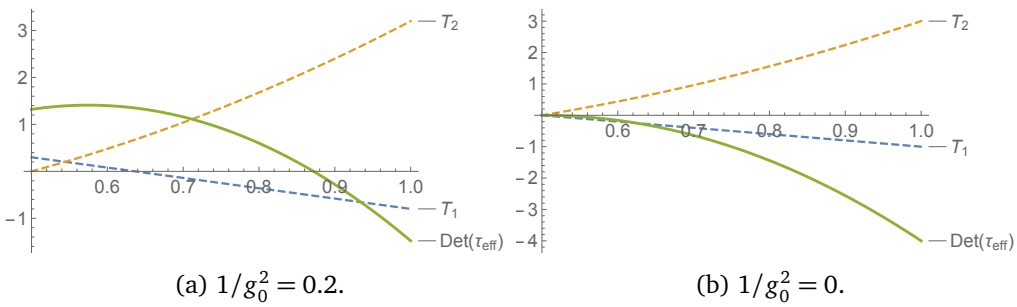

(a) $1/g_0^2 = 0.2$.      (b) $1/g_0^2 = 0$.

Figure 8: $\text{Det}(\tau_{\text{eff}})$ (solid line) and string tensions (dashes lines) of $SU(3)_{-4} + 6\mathbf{F}$ in the first chamber.

and it becomes a one-dimensional real line $\mathcal{C}_{\text{phys}}^{(1)} : 2\phi = \phi_2 \geq 0$ at infinite coupling. In the second chamber, there is no physical domain in the Coulomb branch. The metric on the physical Coulomb branch with $T \geq 0$ is non-negative as depicted in Figure 8.

Instanton operator analysis at 1-instanton level tells us that the global symmetry of this theory will be enhanced to $SO(12) \times U(1)$ at the fixed point. Following [15, 30], one can show that there is a conserved current multiplet in the antisymmetric representation of the classical $SU(6)$ flavor symmetry due to a 1-instanton operator which enhances the global symmetry $U(6) \times U(1)_I$ to $SO(12) \times U(1)$, where $U(1)_I$ is the instanton symmetry. It is not clear, however, whether or not this theory has a 6d uplift in the UV because the enhanced global symmetry is not of affine type.

## 3.2 Additional conjectures

In principle, we can classify all non-trivial 5d gauge theories for simple gauge groups of arbitrary rank using the criterion in (13). For low rank, this is a manageable problem and thus we classify all gauge theories with $r_G \leq 3$ using (13) in the next section. However, our method of classification in this case involves an analysis on the entire Coulomb branch, and thus becomes a difficult task for higher rank gauge theories.

To circumvent this issue, we propose the following three additional conjectures, each of which we have checked leads to the same classification of 5d theories obtained via our main conjecture, up to $r_G \leq 3$ analytically and up to $r_G \leq 5$ numerically:

---

more detail in Section 4.

> **More Conjectures**
>
> 1. If $\tau_{\mathrm{eff}}(\phi) > 0$ *somewhere* in the naive Coulomb branch $\mathcal{C}$ (possibly including unphysical regions), then there exists a physical Coulomb branch $\mathcal{C}_{\mathrm{phys}}$.
>
> 2. If all $T(\phi) > 0$ in some region $\mathcal{C}_{T>0} \subseteq \mathcal{C}$, then $\tau_{\mathrm{eff}}(\phi) > 0$ in $\mathcal{C}_{T>0}$.
>
> 3. If the prepotential $\mathcal{F}(\phi) > 0$ *everywhere* in the the Coulomb branch $\mathcal{C}$ (possibly including unphysical regions), there exists a physical Coulomb branch $\mathcal{C}_{\mathrm{phys}}$.
>
> 4. Conjectures 1-3 above are equivalent to our main conjecture.

Physical or mathematical proofs for the additional conjectures above are currently lacking. However, the checks we did up to $r_G \leq 5$ are very compelling evidences for the above conjectures. So we believe that they also hold for arbitrary higher-rank gauge theories. We leave the proof of the equivalence of these conjectures to future study.

Each of the above three conjectures has its own interesting implications, which we now discuss briefly. Conjecture 1 can be viewed as an extension of the criterion previously discussed in [8]. There, the authors imposed that the constraint of metric positivity everywhere on the Weyl chamber $\mathcal{C}$, whereas Conjecture 1 merely requires that there exist *some* regions in $\mathcal{C}$ for which the metric is positive definite. Stated differently, we do not assume that the physical Coulomb branch $\mathcal{C}_{\mathrm{phys}} = \mathcal{C}$.

Conjecture 2 is quite interesting because in all cases we have checked, the condition that the tensions be positive on some region $\mathcal{C}_{\mathrm{phys}}$ guarantees that the metric is also positive definite on $\mathcal{C}_{\mathrm{phys}}$. Therefore, according to our main conjecture, the existence of such a region $\mathcal{C}_{\mathrm{phys}}$ is sufficient to imply the existence of a non-trivial fixed point. From a practical standpoint, Conjecture 2 is much easier to utilize for a systematic classification of 5d theories because the mathematical conditions we need to impose only involve the gradient of the prepotential, whereas metric positivity requires diagonalization of the Hessian of the prepotential. Moreover, the constraint of positive tension is convenient because we only need to apply it to a local region of the naive Coulomb branch (as opposed to the entire Coulomb branch) to identify a legitimate theory.

Conjecture 3 is essentially motivated by the convergence of the partition function on a compact five-manifold. More precisely, Conjecture 3 asserts that positivity of $\mathcal{F}$ on the entire Coulomb $\mathcal{C}$ branch guarantees the convergence of the naive perturbative partition function. However, it is not clear if Conjecture 3 can guarantee the convergence of the full partition function including non-perturabive contributions. We discuss the details of the partition function on $S^5$ and its connection to Conjecture 3 in Appendix C. One practical advantage of Conjecture 3 is that it only involves checking the prepotential itself, rather than its first or second derivatives. However, this advantage comes at the expense of having to analyze the prepotential on the entire Coulomb branch as opposed to a local region.

We remark that Conjectures 2 and 3 are complementary: Conjecture 2 can be used to argue the existence of non-trivial 5d gauge theories by locally identifying a physical Coulomb branch $\mathcal{C}_{\mathrm{phys}}$, while Conjecture 3 can be used to exclude theories by again locally identifying a region of $\mathcal{C}$ that does not admit a positive prepotential. Therefore, a combination of these two conjectures gives us a powerful method to fully classify 5d SQFTs with simple gauge groups. In the following section we will explicitly describe the classification we obtained by using Conjectures 2 and 3.

# 4 Classification of 5d CFTs

## 4.1 Classification overview

Analysis on the Coulomb branch of the moduli space provides us a systematic way to classify non-trivial 5d gauge theories with interacting fixed points. For the first step toward this direction, we will classify all possible gauge theories with simple gauge group $G$.

An important and very useful assumption in our classification is the idea that the dimension of the largest matter representation must be less than the dimension of the adjoint representation. The reason for this assumption is the fact that the matter hypermultiplets contribute terms that "reduce" the convexity of the perturbative prepotential (1). In order to ensure the existence of a physical Coulomb branch for which the prepotential is convex, it is therefore necessary that the vector multiplet contributions to the prepotential "balance" against the hypermultiplet contributions. Building on an observation made in [8], we translate this requirement into the restriction that the lengths of the weights of a given representation be no larger than those of the adjoint representation. If this were not the case, then even for a single matter hypermultiplet it would be possible to locate a direction in the naive Coulomb branch for which the prepotential is negative[5] violating Conjecture 3.

In the following classification, we will distinguish between two types of theories—*standard* and *exceptional*. Standard theories are classes of 5d gauge theories with classical gauge groups for which the same specifications for the matter content (depending on the rank of the gauge group) applies for all rank $r_G$. The expressions bounding the matter content for standard theories are simple linear expressions involving $r_G$ and the numbers of flavors. By contrast, exceptional theories exist only for some sporadic lower rank cases, where $r_G \leq 8$. For these theories, there is no simple formula constraining the matter content and therefore they must be studied on a case-by-case basis.

In the first sub-section, we will present the classification of all standard theories. We will first choose some particular subregions in the Weyl chamber (where the choices of subregions are motivated by data from lower rank studies) and examine these subregions using Conjecture 3. This will set upper bounds on the matter content for non-trivial QFTs. Note that this does not guarantee the existences of such theories, but merely exclude the theories beyond the bounds. Then we will apply Conjecture 2 along the same subregions to confirm the existence of the theories below these bounds. This will allow us to finish the full classification of non-trivial CFTs with single gauge nodes of higher ranks $r_G > 8$.

In the second sub-section, we will deal with exceptional cases at lower ranks $r_G \leq 8$. Our classification program involves both analytic and numerical search methods. By 'analytic', we mean that we have used a symbolic computing tool to derive precise inequalities describing the physical Coulomb branches with positive metric. By 'numerical', we mean that we have implemented a numerical search on the lattice discretization of a fundamental Weyl chamber to locate a region admitting physical Coulomb branch. Having determined such regions, we then analytically test whether or not such regions admit non-trivial theories using Conjectures 2 and 3.

For $r_G \leq 3$, we employ only our main conjecture to analytically perform the classification, without using the additional conjectures. For $3 < r_G \leq 5$, we employ a numerical search combined with our main conjecture to identify suitable theories, and then use Conjectures 2

---

[5]As described in Appendix A.3, the length $l(e)$ of a root $e$ is defined to be the sum of the coefficients of $e$ in a basis of simple roots, and similarly for a weight $w$. We note that it is always possible to locate a region $\phi^* \in \mathcal{C}$ for which $e \cdot \phi^* = l(e)$ and $w \cdot \phi^* = l(w)$, and hence at infinite coupling $1/g_0^2 = 0$, the prepotential is proportional to $6\mathcal{F}(\phi^*) = \sum_{e \in \text{roots}} |l(e)|^3 - \sum_{w \in \mathbf{R}_f} |l(w)|^3$, and the classical CS term $\sum_{w \in \mathbf{F}} (w \cdot \phi^*)^3 = \sum_{w \in \mathbf{F}} l(w)^3 = 0$. It is then evident that $\mathcal{F}(\phi^*)$ will be negative if the quadratic Dynkin index of $\mathbf{R}_f$ is larger than the quadratic Dynkin index of the adjoint representation—see Appendix B for further discussion.

and 3 to confirm the existence of these theories analytically. For $5 < r_G \leq 8$, we numerically perform the classification by using Conjectures 2 and 3.

**Marginal theories vs. 5d CFTs**  One additional idea essential to our classification is the notion of a *marginal* theory. Marginal theories are those theories for which there exists a physical Coulomb branch $\mathcal{C}_{\text{phys}}$ with positive metric, but the strong coupling limit $1/g_0^2 \to 0$ is not a 5d SCFT fixed point. One distinguished characteristic of marginal theories is the vanishing of the some of the eigenvalues of $\tau_{\text{eff}}$ at infinite coupling limit, i.e. the metric is degenerate along $\mathcal{C}_{\text{phys}}$. In addition, the physical Coulomb branch with positive string tensions collapses to a lower dimensional space at infinite coupling limit:

$$\dim \mathcal{C}_{\text{phys}}\big|_{g_0^2 \to \infty} < r_G \,. \tag{17}$$

In order to preserve the full dimensionality of $\mathcal{C}_{\text{phys}}$, it is necessary to keep the gauge coupling finite, i.e. $g_0^2 < \infty$. Finite gauge coupling introduces a scale to such theories, implying that their UV completions cannot be 5d SCFTs. Some of these theories will have 6d UV completions instead of 5d SCFT completions. We will discuss such theories uplifted to 6d theories in Section 5. For other marginal theories, we do not have a clear understanding of their UV completions, or if they even exist.

We use the term *descendants* to refer to theories that can be obtained from some other theory with a larger number of matter hypermultiplets via RG flow after integrating out some massive matter hypermultiplets. All the descendants of the marginal theories flow to 5d SCFT fixed points at infinite coupling.

The reason for this, as explained in [8], is that the cubic deformations to the prepotential (coming from the terms $|w \cdot \phi + m_f|^3$) resulting from sending a mass parameter $m_f \to \pm\infty$ and integrating out the matter make the prepotential "more convex". The only other deformations to the prepotential arising from integrating out $N_{\mathbf{R}_f}$ massive matter hypermultiplets in the representation $\mathbf{R}_f$ with mass terms $\text{sgn}(m_f)|m_f|$ are terms of the form $-\text{sgn}(m_f)N_{\mathbf{R}_f} c_{\mathbf{R}_f}^{(3)}/2$ contributing to the classical CS level $\kappa$ [8]. Therefore, for $SU(N \geq 3)$ theories, one can trade matter hypermultiplets for the CS level $\kappa$, and such theories will have interacting fixed points as demonstrated by our classification. On the other hand, if one increases $\kappa$ or adds more matter to these marginal theories described above, the resulting theories have no physical Coulomb branch and consequently they are trivial.

**Notational conventions**  In the following discussion, we adopt the condensed notation

$$G_\kappa + \sum_f N_{\mathbf{R}_f} \mathbf{R}_f \,, \tag{18}$$

to refer to a gauge theory with simple gauge group $G$ with CS level $\kappa$ coupled to $N_{\mathbf{R}_f}$ matter hypermultiplets (or half-hypermultiplets) transforming in the representation $\mathbf{R}_f$. We encounter the following representations:

- **F**, fundamental

- **Sym**, rank two symmetric

- **AS**, rank-2 antisymmetric

- **TAS**, rank-3 antisymmetric

- **S**, spinor

- **C**, conjugate spinor.

## 4.2 Standard theories

A class of gauge theories with gauge groups $G = SU(N), Sp(N), SO(N)$ which exist even at large rank can be fully classified by employing our conjectures. We call such theories as *standard* theories as mentioned above. The results are summarized in Table 1. The theories in these tables are marginal theories. We expect that all of the standard have 6d uplifts at strong coupling, a point that we discuss further in Section 5. The descendants of these marginal theories are expected to have non-trivial 5d SCFT fixed points at UV. $\mathcal{C}_{\text{phys}}$ is the subregion in the Weyl chamber within which we have confirmed the metric and the string tensions of the corresponding theory are both positive.

$F$ is the global symmetry at the UV fixed point which we obtain by using 1-instanton analysis in [15,16,26,30].[6] Some lower rank theories have further symmetry enhancements which are listed in Table 2. The full global symmetry could be in general different from those listed in the table if the symmetry enhancement occurs at two or higher instanton level. In some cases, the global symmetry group is affinized by instanton operators. This may signal that the theory is uplifted to a 6d theory at the UV fixed point. Some of these global symmetries have already been confirmed by identifying brane constructions or explicit 6d uplifts of these theories [21,22,31].

In the following section we present an algorithm for the classification of the standard theories listed in Table 1.

### 4.2.1 $SU(N)$ gauge theories

Let us start with the classification of standard $SU(N)$ gauge theories with $N_{\text{F}}$ fundamental, $N_{\text{Sym}}$ symmetric, and $N_{\text{AS}}$ antisymmetric hypermultiplets. Note that other representations are not allowed for higher rank theories $r_G > 8$. For lower rank theories $r_G \leq 8$, we can also add $N_{\text{TAS}}$ matters in the rank-3 antisymmetric representation. But theories with $N_{\text{TAS}} > 0$ are exceptional theories that we will discuss separately below. Matter hypermultiplets in other representations are not allowed.

In the following discussion, we express the roots and weights in the orthogonal basis. The prepotential for $SU(N)_\kappa + N_{\text{F}}\mathbf{F} + N_{\text{AS}}\mathbf{AS}$ in the Weyl chamber $a_1 \geq a_2 \geq \cdots \geq a_{N-1} \geq 0$ is given by

$$\mathcal{F}_{SU(N)} = \frac{1}{2g_0^2}\sum_{i=1}^{N}a_i^2 + \frac{\kappa}{6}\sum_{i=1}^{N}a_i^3 + \frac{1}{6}\sum_{i<j}^{N}(a_i-a_j)^3 - \frac{N_{\text{F}}}{12}\sum_{i=1}^{N}|a_i|^3 - \frac{N_{\text{AS}}}{6}\sum_{i<j}^{N}|a_i+a_j|^3, \quad (19)$$

with $a_N = -\sum_i a_i$. A symmetric matter hypermultiplet has the same contribution to the prepotential as the contribution from $(N_{\text{AS}} = 1, N_{\text{F}} = 8)$ hypermultiplets. It follows that $(N_{\text{AS}} = 1, N_{\text{F}} = 8)$ hypermultiplets can be traded with $N_{\text{Sym}} = 1$ at the level of the prepotential. So we will consider the cases $N_{\text{Sym}} = 0$ and retrive the cases with $N_{\text{Sym}} > 0$ by using the "equivalence" $(N_{\text{AS}} = 1, N_{\text{F}} = 8) \sim N_{\text{Sym}} = 1$ as needed.

We will examine the following three regions of the Weyl chamber,

$$\mathcal{S}_{SU(N)}^{(1)} = (a, a, \cdots, a, -(N-1)a), \quad \mathcal{S}_{SU(N)}^{(2)} = (a, 0, \cdots, 0, -a), \quad \mathcal{S}_{SU(N)}^{(3)} = (a, a, 0, \cdots, 0, -2a), \quad (20)$$

Firstly, Conjecture 3 imposes the condition for non-trivial $SU(N)$ theories that the prepotential should be positive $\mathcal{F}_{SU(N)} > 0$ along these three regions. Then each of three regions gives rise

---

[6]In some cases, like $Sp(N)$ with fundamental matter, the 1-instanton analysis was supplemented by additional data. Also there are several subtleties and technical difficulties involved in the 1-instanton analysis, which in some cases prevent us from carrying the calculation. The cases of $SU(N)$ groups with symmetric matter hypermultiplets are particularly subtle. The symmetries written in these cases are the minimal ones we are sure of, but there may well be additional enhancements by 1-instanton states for low $N$ cases.

Table 1: Classes of standard gauge theories. Note that all of these theories are marginal and their descendants have 5d conformal fixed points in the UV. See (20), (35).

(a) Marginal $SU(N)$ theories with CS level $\kappa$, $N_{\text{Sym}}$ symmetric, $N_{\text{AS}}$ antisymmetric, and $N_{\text{F}}$ fundamental hypermultiplets. See (20) for an explicit description of the physical Coulomb branches described in the far right column.

| $N_{\text{Sym}}$ | $N_{\text{AS}}$ | $N_{\text{F}}$ | $|\kappa|$ | $F$ | $\mathcal{C}_{\text{phys}}$ |
|---|---|---|---|---|---|
| 1 | 1 | 0 | 0 | $A_1^{(1)} \times U(1)$ | $\mathcal{S}_{SU(N)}^{(2)}$ |
| 1 | 0 | $N-2$ | 0 | $A_{N-2}^{(1)} \times U(1)$ | $\mathcal{S}_{SU(N)}^{(2)}$ |
| 1 | 0 | 0 | $\frac{N}{2}$ | $U(1)^2$ | $\mathcal{S}_{SU(N)}^{(1)}$ |
| 0 | 2 | 8 | 0 | Even $N$ : $E_7^{(1)} \times A_1^{(1)} \times A_1^{(1)} \times SU(2)$ <br> Odd $N$ : $D_8^{(1)} \times A_1^{(1)} \times SU(2)$ | $\mathcal{S}_{SU(N)}^{(2)}$ |
| 0 | 2 | 7 | $\frac{3}{2}$ | Even $N$ : $D_8^{(1)} \times SU(2)$ <br> Odd $N$ : $E_7^{(1)} \times SU(2) \times SU(2)$ | $\mathcal{S}_{SU(N)}^{(3)}$ |
| 0 | 1 | $N+6$ | 0 | $A_{N+6}^{(1)} \times U(1)$ | $\mathcal{S}_{SU(N)}^{(2)}$ |
| 0 | 1 | 8 | $\frac{N}{2}$ | $SO(16) \times U(1)^2$ | $\mathcal{S}_{SU(N)}^{(1)}$ |
| 0 | 0 | $2N+4$ | 0 | $D_{2N+4}^{(1)}$ | $\mathcal{S}_{SU(N)}^{(2)}$ |

(b) Marginal $Sp(N)$ theories with $N_{\text{AS}}$ antisymmetric and $N_{\text{F}}$ fundamental hypermultiplets. See (35) for a description of the physical Coulomb branches.

| $N_{\text{AS}}$ | $N_{\text{F}}$ | $F$ | $\mathcal{C}_{\text{phys}}$ |
|---|---|---|---|
| 1 | 8 | $E_8^{(1)} \times SU(2)$ | $\mathcal{S}_{Sp(N)}$ |
| 0 | $2N+6$ | $D_{2N+6}^{(1)}$ | $\mathcal{S}_{Sp(N)}$ |

(c) Marginal $SO(N > 4)$ theories with $N_{\text{F}}$ fundamental hypermultiplets. See (42) for a description of the physical Coulomb branch.

| $N_{\text{F}}$ | $F$ | $\mathcal{C}_{\text{phys}}$ |
|---|---|---|
| $N-2$ | $A_{2N-5}^{(2)}$ | $\mathcal{S}_{SO(N)}$ |

Table 2: $SU(N)$ theories with further symmetry enhancement.

| $N$ | $N_{\text{AS}}$ | $N_{\text{F}}$ | $|\kappa|$ | $F$ |
|---|---|---|---|---|
| 4 | 2 | 8 | 0 | $E_7^{(1)} \times B_3^{(1)}$ |
| 5 | 2 | 7 | $\frac{3}{2}$ | $E_7^{(1)} \times G_2$ |
| 4 | 2 | 7 | $\frac{3}{2}$ | $B_9^{(1)}$ |
| 5 | 1 | 11 | 0 | $A_{11}^{(1)} \times A_1^{(1)}$ |
| 4 | 1 | 10 | 0 | $A_{11}^{(1)}$ |
| 6 | 1 | 8 | 3 | $SO(16) \times SU(2) \times U(1)$ |
| 5 | 1 | 8 | $\frac{5}{2}$ | $E_8^{(1)} \times U(1)$ |
| 4 | 1 | 8 | 2 | $E_8^{(1)} \times U(1)$ |

to the following three inequalities,

$$
\begin{aligned}
\mathcal{S}^{(1)}_{SU(N)} : \quad & N_{\mathbf{F}}(N^2 - 2N + 2) + 2|\kappa|N(N-2) + N_{\mathbf{AS}}(N-2)(N^2 - 4N + 8) \le 2N^3, \\
\mathcal{S}^{(2)}_{SU(N)} : \quad & N_{\mathbf{F}} + (N-2)N_{\mathbf{AS}} \le 2N + 4, \\
\mathcal{S}^{(3)}_{SU(N)} : \quad & 5N_{\mathbf{F}} + 5N_{\mathbf{AS}}(N-2) + 6|\kappa| \le 10N + 24.
\end{aligned}
\tag{21}
$$

For a given $N$, these inequalities set some upper bounds on the matter content of the theories with non-trivial fixed points. We may be able to find other independent inequalities by examining other regions in the Weyl chamber. But we will show shortly that these inequalities are enough for higher rank theories. By analyzing these three inequalities, it can be shown that for $N > 6$ there are only five cases that saturate one of these inequalities while not violating the others:

$$
(N_{\mathbf{AS}}, N_{\mathbf{F}}, |\kappa|) = (2, 8, 0), \quad (2, 7, \tfrac{3}{2}), \quad (1, N+6, 0), \quad (1, 8, \tfrac{N}{2}), \quad (0, 2N+4, 0). \tag{22}
$$

Other theories with $N > 6$ having more matter or higher Chern-Simons levels are all excluded by Conjecture 3 since they have negative prepotential along one of the above three regions.

We will now confirm that the upper boundary theories in (22) are indeed non-tivial theories having interacting UV fixed points by testing Conjecture 2 along the three subregions $\mathcal{S}^{(i=1,2,3)}_{SU(N)}$. First, we expand the Coulomb branch parameters around $\mathcal{S}^{(1)}_{SU(N)}$ as

$$
a_i = a + \delta_i, \quad \delta_i \ll 1, \tag{23}
$$

with $\delta_i > \delta_{i+1}$. Then the string tensions can be approximated as

$$
\begin{aligned}
T_i &= (\partial_{a_i} - \partial_{a_{i+1}})\mathcal{F}_{SU(N)} \\
&= \left( \frac{1}{g_0^2} + \left( N + \kappa - \frac{N_{\mathbf{F}}}{2} - \frac{3}{2}NN_{\mathbf{AS}} + 4N_{\mathbf{AS}} \right) a \right)(\delta_i - \delta_{i+1}) + \mathcal{O}(\delta^2), \\
T_{N-1} &= \partial_{a_{N-1}}\mathcal{F}_{SU(N)} \\
&= \frac{N}{g_0^2}a + \left( N^3 - \kappa N(N-2) - \frac{N_{\mathbf{F}}}{2}(N^2 - 2N + 2) - \frac{N_{\mathbf{AS}}}{4}(N^3 - 6N^2 + 16N - 16) \right)\frac{a^2}{2} + \mathcal{O}(\delta).
\end{aligned}
\tag{24}
$$

It is obvious that all the string tensions are positive at weak coupling $1/g_0^2 \gg 1$ since $\delta_i > \delta_{i+1}$. However, in the strong coupling limit $1/g_0^2 \to 0$, all of the string tensions are positive, i.e. $T > 0$, only if the matter content of the theory satisfies the following inequalities:

$$
\begin{aligned}
& 2N + 2\kappa - N_{\mathbf{F}} - N_{\mathbf{AS}}(3N - 8) \ge 0, \\
& N^3 - \kappa N(N-2) - \frac{N_{\mathbf{F}}}{2}(N^2 - 2N + 2) - \frac{N_{\mathbf{AS}}}{4}(N^3 - 6N^2 + 16N - 16) \ge 0.
\end{aligned}
\tag{25}
$$

When all the string tensions are positive, we identify the corresponding theory as a good theory with an interacting UV fixed point by using Conjecture 2.

Similarly, it is straightforward to show that the string tension analysis around $\mathcal{S}^{(2)}_{SU(N)}$ leads to the inequalities

$$
2N + 4 - N_{\mathbf{F}} - N_{\mathbf{AS}}(N-2) \ge 0, \quad N_{\mathbf{AS}} \le 2. \tag{26}
$$

Lastly, the string tensions near $\mathcal{S}^{(3)}_{SU(N)}$ are approximated as

$$
\begin{aligned}
T_1 &= \left( \frac{1}{g_0^2} + \left( N + k - \frac{1}{2}N_{\mathbf{F}} - \frac{1}{2}N_{\mathbf{AS}}(N-2) \right) a \right) (\delta_1 - \delta_2) + \mathcal{O}(\delta^2), \\
T_2 &= \frac{a}{g_0^2} + \left( N + 4 + k - \frac{1}{2}N_{\mathbf{F}} - \frac{1}{2}N_{\mathbf{AS}}(N+2) \right) a^2/2 + \mathcal{O}(\delta), \\
T_i &= \left( \frac{1}{g_0^2} + (4 - 2N_{\mathbf{AS}})a \right) (\delta_i - \delta_{i+1}) + \mathcal{O}(\delta), \\
T_{N-1} &= \frac{2a}{g_0^2} + 2 \left( N + 2 - k - \frac{1}{2}N_{\mathbf{F}} - \frac{1}{2}N_{\mathbf{AS}}(N-3) \right) a^2 + \mathcal{O}(\delta),
\end{aligned}
\tag{27}
$$

where $a_1 = a + \delta_1, a_2 = a + \delta_2$ and $a_i = \delta_i$ for $3 \leq i \leq N-1$ with infinitesimal parameters $\delta_i$. Therefore the analysis $T > 0$ around $S^{(3)}_{SU(N)}$ gives the inequalities

$$
N_{\mathbf{AS}} \leq 2 \,, \quad 2N + 2k - N_{\mathbf{F}} - N_{\mathbf{AS}}(N-2) \geq 0, \quad 2N + 4 - 2\kappa - N_{\mathbf{F}} - N_{\mathbf{AS}}(N-3) \geq 0. \tag{28}
$$

We now have three sets of inequalities (25), (26), (26). Conjecture 2 tells us that if an $SU(N)$ theory on one of the three subregions $\mathcal{S}^{(i=1,2,3)}_{SU(N)}$ satisfies the corresponding inequality, the theory is a non-trivial theory having interacting fixed point at UV. With this criterion, one can easily show that all of the upper boundary theories in (22) are non-trivial theories. We will call these theories, and their descendants obtained by integrating out massive matter, standard $SU(N)$ gauge theories. The standard $SU(N)$ theories are summarized in Table 1a. Note that we obtained results for the theories with symmetric hypermultiplets by trading $(N_{\mathbf{AS}}, N_{\mathbf{F}}) = (1, 8) \rightarrow N_{\mathbf{Sym}} = 1$. The theories (and their descendants) in Table 1a exist at any $N$ and they are the only non-trivial theories at $N \geq 9$, which is proven above relying on Conjecture 2 and 3. This completes the full classification of standard $SU(N)$ gauge theories. In fact, all the theories listed in Table 1a lift to 6d SCFTs in the UV, and hence they are marginal theories. We discuss these theories and their 6d uplifts in more detail in Section 5.

### 4.2.2 $Sp(N)$ gauge theories

Let us turn to $Sp(N)$ gauge theories. $Sp(N)$ gauge theory can have $N_{\mathbf{F}}$ fundamental, $N_{\mathbf{AS}}$ antisymmetric, and $N_{\mathbf{TAS}}$ rank-3 antisymmetric representations of $Sp(N)$. We claim that no other representations are permitted, and furthermore $N_{\mathbf{TAS}} > 0$ only for theories with $N \leq 4$. The theories with $N_{\mathbf{TAS}} > 0$ are exceptional theories, and will be discussed separately below. In this section, we consider the standard theories $Sp(N) + N_{\mathbf{F}}\mathbf{F} + N_{\mathbf{AS}}\mathbf{AS}$. The $Sp(N)$ theory has a single Weyl subchamber $a_1 \geq a_2 \geq \cdots \geq a_N \geq 0$. The prepotential is given by

$$
\mathcal{F}_{Sp(N)} = \frac{1}{2g_0^2} \sum_{i=1}^{N} a_i^2 + \frac{1}{6} \left( \sum_{i<j}^{N} \left( (a_i - a_j)^3 + (a_i + a_j)^3 \right) (1 - N_{\mathbf{AS}}) + (8 - N_{\mathbf{F}}) \sum_{i=1}^{N} a_i^3 \right). \tag{29}
$$

Since the $Sp(N)$ theory has a single Weyl chamber, in sharp contrast to the $SU(N)$ cases with many distinct chambers, we can in fact examine every point on the Weyl chamber without restricting to particular subregions and thereby obtain a full classification using only Conjecture 2.

The string tensions are

$$
\begin{aligned}
T_i &= (\partial_{a_i} - \partial_{a_{i+1}})\mathcal{F}_{Sp(N)} \\
&= (a_i - a_{i+1})\left( \frac{1}{g_0^2} + \sum_{j=1}^{i}(2a_j - a_i - a_{i+1})(1-N_{\mathbf{AS}}) + \frac{1}{2}((2N-4)(1-N_{\mathbf{AS}}) + 8 - N_{\mathbf{F}})(a_i + a_{i+1}) \right),
\end{aligned}
$$

$$
T_N = \partial_{a_N}\mathcal{F}_{Sp(N)} = a_N\left( \frac{1}{g_0^2} + \frac{8-N_{\mathbf{F}}}{2}a_N + 2(1-N_{\mathbf{AS}})a_T \right), \tag{30}
$$

where $a_T = \sum_{i=1}^{N-1} a_i$. The condition on the last string tension $T_N > 0$ gives the inequality

$$
N_{\mathbf{AS}} \leq \frac{8-N_{\mathbf{F}}}{4}\frac{a_N}{a_T} + 1. \tag{31}
$$

The right hand side of this equation is maximized near $a_1 = a_2 = \cdots = a_N$ and the result is

$$
N_{\mathbf{AS}} \leq \frac{4N - N_{\mathbf{F}} + 4}{4(N-1)}. \tag{32}
$$

For the theories with $N \geq 5$, this condition allows only the cases with $N_{\mathbf{AS}} = 0, 1$.

When $N_{\mathbf{AS}} = 1$, the tension positivity conditions reduce to the single condition

$$
N_{\mathbf{F}} \leq 8 \quad (\text{when } N_{\mathbf{AS}} = 1). \tag{33}
$$

When $N_{\mathbf{AS}} = 0$, the string tensions can be written as

$$
T_i = (a_i - a_{i+1})\left( \frac{1}{g_0^2} + 2\sum_{j=1}^{i-1}a_j + \frac{2N+8-N_{\mathbf{F}}-2i}{2}a_i^2 + \frac{2N+4-N_{\mathbf{F}}-2i}{2}a_{i+1}^2 \right),
$$

$$
T_N = a_N\left( \frac{1}{g_0^2} + \frac{8-N_{\mathbf{F}}}{2}a_N + 2a_T \right). \tag{34}
$$

The positive tension conditions $T > 0$ admit the maximum number of matter hypermultiplets near the subregion

$$
\mathcal{S}_{Sp(N)} = (a, 0, \cdots, 0). \tag{35}
$$

Around $\mathcal{S}_{Sp(N)}$, the only non-trivial condition is $T_1 > 0$ which gives rise to the bound

$$
N_{\mathbf{F}} \leq 2N + 6. \tag{36}
$$

The standard $Sp(N)$ theories are those summarized in Table 1b along with their descendants. The theories in Table 1b have 6d uplifts at the UV fixed point, so they are marginal theories. Their descendants are expected to have 5d CFT fixed points in UV. Note that, when $N_{\mathbf{F}} = 0$, the $Sp(N)$ gauge theory has a discrete theta angle $\theta = 0, \pi$ associated to $\pi(Sp(N)) \cong \mathbb{Z}_2$. Thus there are two more non-trivial $Sp(N)$ theories with $(N_{\mathbf{AS}} = 0, N_{\mathbf{F}} = 0)$ and $(N_{\mathbf{AS}} = 1, N_{\mathbf{F}} = 0)$.

One can immediately show that, for these standard theories, the prepotential (29) is always positive within the Weyl chamber. If we add more matter to these theories, their prepotential will become negative somewhere in the Weyl chamber. This confirms Conjecture 3 for the standard $Sp(N)$ gauge theories.

### 4.2.3 $SO(N)$ gauge theories

We consider the case of $SO(N)$ with $N_{\mathbf{F}}$ fundamental flavors. Spinor hypermultiplets can also be added when $N \leq 14$, but because such theories are not standard, we discuss them in the next subsection. In the case $N = 2r + 1$, there is only a single Weyl subchamber to consider, namely $a_1 \geq a_2 \geq \cdots \geq a_r \geq 0$. The prepotential for $SO(2r+1) + N_{\mathbf{F}}\mathbf{F}$ is

$$\mathcal{F}_{SO(2r+1)} = \frac{1}{g_0^2}\sum_{i=1}^{r}a_i^2 + \frac{1}{6}\sum_{i<j}^{r}[(a_i+a_j)^3 + (a_i-a_j)^3] + \frac{1-N_{\mathbf{F}}}{6}\sum_{i=1}^{r}a_i^3. \tag{37}$$

Using the above formula for the prepotential, we see that the tensions are given by:

$$
\begin{aligned}
T_{i<r} &= \left(\partial_{a_i} - \partial_{a_{i+1}}\right)\mathcal{F}_{SO(2r+1)} \\
&= \frac{2}{g_0^2}(a_i - a_{i+1}) + 2(a_i - a_{i+1})\sum_{j=1}^{i-1}a_j \\
&\quad + a_{i+1}^2\left(1 - \frac{1}{2}(N - N_{\mathbf{F}} - 2i - 2)\right) + \left(\frac{a_i}{2}(N - N_{\mathbf{F}} - 2i) - 2a_{i+1}\right),
\end{aligned}
\tag{38}
$$

$$T_r = 2\partial_{a_r}\mathcal{F}_{SO(2r+1)} = \frac{2}{g_0^2}a_r + 4a_r\sum_{j=1}^{r-1}a_j + (1-N_{\mathbf{F}})a_r^2. \tag{39}$$

Let us first focus on the expression for $T_1$:

$$T_1 = \frac{2}{g_0^2}(a_1 - a_2) + \frac{1}{2}(a_1 - a_2)((N - N_{\mathbf{F}} - 2)a_1 + (N - N_{\mathbf{F}} - 6)a_2). \tag{40}$$

It is possible to maximize the number of fundamental matters while keeping $T_1$ positive by setting $a_2 = 0$ in the infinite coupling regime $1/g_0^2 \ll 1$:

$$N_{\mathbf{F}} \leq N - 2. \tag{41}$$

Notice that when $a_2 = 0$, the remaining Coulomb branch parameters $a_j = 0$ for $j > 2$ according to the definition of the fundamental Weyl chamber. Therefore, we see that we can maximize $N_{\mathbf{F}}$ while keeping all tensions positive along the locus

$$\mathcal{S}_{SO(2n+1)} = (a, 0, \ldots, 0). \tag{42}$$

For $N_{\mathbf{F}} = N - 2$, the only way to maintain positive tension is to keep the gauge coupling finite, $g_0^2 < \infty$. The resulting theory is a marginal theory with a 6d uplift at the UV fixed point. When the number of fundamental matters $N_{\mathbf{F}} > N - 2$, there is no physical Coulomb branch at infinite coupling $1/g_0^2 = 0$ with positive string tensions, and therefore such theories are trivial.

For $SO(2r)$ gauge theories with fundamental matter, the proof works identically to the above case, as we will now show. The prepotential for the $SO(2r)$ theory is

$$\mathcal{F}_{SO(2r)} = \frac{1}{g_0^2}\sum_{i=1}^{r}a_i^2 + \frac{1}{6}\sum_{i<j}^{r}[(a_i+a_j)^3 + (a_i-a_j)^3] - \frac{N_{\mathbf{F}}}{6}\sum_{i=1}^{r}a_i^3. \tag{43}$$

We again parametrize the fundamental Weyl chamber by the condition $a_1 \geq a_2 \geq \cdots \geq a_r \geq 0$. By computing derivatives of the above formula with respect to $\phi_i$, we find that the monopole

tensions are given by

$$
\begin{aligned}
T_{i<r} &= \left(\partial_{a_i} - \partial_{a_{i+1}}\right)\mathcal{F}_{SO(2r)} \\
&= \frac{2}{g_0^2}(a_i - a_{i+1}) + 2(a_i - a_{i+1})\sum_{j=1}^{i-1} a_j \\
&\quad + a_{i+1}^2\left(1 - \frac{1}{2}(N - N_{\mathbf{F}} - 2i - 2)\right) + \left(\frac{a_i}{2}(N - N_{\mathbf{F}} - 2i) - 2a_{i+1}\right),
\end{aligned}
\tag{44}
$$

$$
\begin{aligned}
T_r &= (\partial_{a_{r-1}} + \partial_{a_r})\mathcal{F}_{SO(2r)} \\
&= \frac{2}{g_0^2}(a_{r-1} + a_r) + 2(a_{r-1} + a_r)\sum_{j=1}^{r-2} a_j \\
&\quad + a_{r-1}\left(2a_r + \frac{a_{r-1}}{2}(N - N_{\mathbf{F}} - 2r + 2)\right) + a_r^2\left(1 - \frac{1}{2}N_{\mathbf{F}}\right).
\end{aligned}
\tag{45}
$$

From the above expressions, we see that the only difference with the tensions $T_i$ associated to the $SO(2r+1)$ gauge theory is in the expression for $T_r$. By arguments identical to the above case $G = SO(2r+1)$, we find again that the bound on the number of fundamental flavors is $N_{\mathbf{F}} \leq N - 2$, with the case $N_{\mathbf{F}} = N - 2$ corresponding to a marginal theory.

## 4.3 Exceptional theories with rank $\leq 8$

**Rank 2**

In the previous section, we confirmed that some 5d theories which go beyond the constraints introduced in [8] are in fact consistent theories with interacting fixed points. We now turn our attention to the case of simple gauge groups $G$ with $r_G = 2$, namely $G = SU(3), Sp(2), G_2$.

We begin by classifying non-trivial $SU(3)$ gauge theories. As discussed in the previous section, $SU(3)_\kappa + N_{\mathbf{F}}\mathbf{F} + N_{\mathbf{Sym}}\mathbf{Sym}$ has two Weyl subchambers, assuming the presence of fundamental hypermultiplets. Applying our main conjecture to the Coulomb branch, we find the list of marginal $SU(3)$ gauge theories displayed in Table 3. On the other hand, all the descendants of these theories obtained by integrating out matter hypermultiplets have non-trivial 5d conformal fixed points. This full list includes all known non-trivial 5d $SU(3)$ gauge theories, $SU(3)_0 + 1\mathbf{F} + 1\mathbf{Sym}$, and $SU(3)_0 + 10\mathbf{F}$, and their descendants. Other theories in the list are new non-trivial $SU(3)$ theories we predict based on new criteria.

We now turn to $Sp(2)$ gauge theories. $Sp(2)$ gauge theories can only have fundamental and antisymmetric matter, again because the dimensions of other representations are too large to admit a positive semi-definite metric on the physical Coulomb branch. There is a single Weyl chamber $\phi_2 \geq \phi_1 \geq \phi_2/2 \geq 0$ in the Dynkin basis. We find the marginal $Sp(2)$ gauge theories listed in the Table 4. The global symmetry algebras of these theories are all affine type [26]. Here $A_N^{(2)}$ denote the affine algebras with outer automorphism twists discussed in [29]. This means that all of them have 6d uplifts at their UV fixed points and their descendants have 5d CFT fixed points. This list completely agrees with matter bounds for $Sp(2)$ gauge theories given in [26]. $Sp(2) + 1\mathbf{AS} + 8\mathbf{F}$ has a two-dimensional Coulomb branch at infinite coupling. This implies that this theory has a two-dimensional tensor branch in 6d. Indeed, this theory is the 6d rank-2 E-string theory on a circle.

Lastly, the $G_2$ gauge theory can contain only fundamental hypermultiplets. This theory has a single Weyl subchamber $2\phi_1 \geq \phi_2 \geq 3\phi_1/2 \geq 0$ in the Dynkin basis. We find that these theories have a non-trivial physical Coulomb branch when $N_{\mathbf{F}} \leq 6$. The theory at $N_{\mathbf{F}} = 6$ has an affine type global symmetry algebra and thus it will be uplifted to 6d at the UV fixed point [26].

Table 3: Marginal theories $SU(3)$ with CS level $\kappa$, $N_{\mathbf{Sym}}$ symmetric and $N_{\mathbf{F}}$ fundamental hypermultiplets. The theories with $\kappa < 0$ can be obtained by the interchange $\phi_1 \leftrightarrow \phi_2$.

| $N_{\mathbf{Sym}}$ | $N_{\mathbf{F}}$ | $\kappa$ | $F$ | $\mathcal{C}_{\text{phys}}$ |
|---|---|---|---|---|
| 1 | 0 | $\frac{3}{2}$ | $SU(2) \times U(1)$ | $2\phi_1 = \phi_2 \geq 0$ |
| 1 | 1 | 0 | $A_1^{(1)} \times U(1)$ | $\phi_1 = \phi_2 \geq 0$ |
| 0 | 10 | 0 | $D_{10}^{(1)}$ | $\phi_1 = \phi_2 \geq 0$ |
| 0 | 9 | $\frac{3}{2}$ | $E_8^{(1)} \times SU(2)$ | $2\phi_1 = \phi_2 \geq 0$ |
| 0 | 6 | 4 | $SO(12) \times U(1)$ | $2\phi_1 = \phi_2 \geq 0$ |
| 0 | 3 | $\frac{13}{2}$ | $U(3) \times U(1)$ | $2\phi_1 = \phi_2 \geq 0$ |
| 0 | 0 | 9 | $U(1)$ | $2\phi_1 = \phi_2 \geq 0$ |

Table 4: Marginal $Sp(2)$ gauge theories with $N_{\mathbf{AS}}$ antisymmetric, $N_{\mathbf{F}}$ fundamental matters.

| $N_{\mathbf{AS}}$ | $N_{\mathbf{F}}$ | $F$ | $\mathcal{C}_{\text{phys}}$ |
|---|---|---|---|
| 3 | 0 | $A_5^{(2)}$ at $\theta = 0$, $Sp(3) \times U(1)$ at $\theta = \pi$ | $2\phi_1 = \phi_2 \geq 0$ |
| 2 | 4 | $A_{11}^{(2)}$ | $2\phi_1 = \phi_2 \geq 0$ |
| 1 | 8 | $E_8^{(1)} \times SU(2)$ | $\phi_1, \phi_2 \geq 0$ |
| 0 | 10 | $D_{10}^{(1)}$ | $\phi_1 = \phi_2 \geq 0$ |

**Rank 3**

Let us start with the $SU(4)$ gauge theories. An $SU(4)$ gauge theory can have matter hypermultiplets in the fundamental, the antisymmetric, and the symmetric representations. When the theory contains symmetric or antisymmetric hypermultiplets, there are four distinguished Weyl chambers. We examine all these chambers and find the list of marginal $SU(4)$ theories in Table 6. Again, this list covers all previously known $SU(4)$ gauge theories in [8, 13–16, 21] and additionally it predicts many new interacting 5d gauge theories. The theories with $N_{\mathbf{AS}} = 4, N_{\mathbf{Sym}} = N_{\mathbf{F}} = 0$ and $\kappa = 0, 1, 2, 3, 4$ are all marginal theories which cannot be obtained from other theories by integrating out massive matter.

Similarly, we can classify all non-trivial $Sp(3)$ gauge theories. The list of marginal $Sp(3)$ theories are given in Table 7. We can also have half-hypermultiplets in the rank-3 antisymmetric representation. However, due to the global anomaly described by Witten in [32], consistent $Sp(3)$ gauge theories with rank-3 antisymmetric matter must also have an odd number of half-hypermultiplets in the fundamental representation. The new criteria confirms that all previously known $Sp(3)$ gauge theories in [8, 26] are physical, with one exception. In [26], $Sp(3) + \frac{1}{2}\mathbf{TAS} + \frac{7}{2}\mathbf{F} + \mathbf{AS}$ may have a fixed point of 6d SCFT on a circle. However, we find that this theory has no physical Coulomb branch. So our classification rules out this theory. It would be interesting to confirm this result using other arguments.

Finally, we come to the case of 5d $SO(7)$ gauge theories, which can have matter hypermul-

Table 5: A marginal $G_2$ gauge theory with $N_F$ fundamental matters.

| $N_F$ | $F$ | $\mathcal{C}_{\text{phys}}$ |
|---|---|---|
| 6 | $A_{11}^{(2)}$ | $2\phi_1 = \phi_2 \geq 0$ |

tiplets in fundamental and spinor representations. The marginal theories $SO(7) + N_F\mathbf{F} + N_S\mathbf{S}$ are listed in Table 8.

**Rank 4**

In this section, we discuss exceptional 5d gauge theories, with gauge algebras $SO(8)$, $SO(9)$, $F_4$, and $SU(5)$. To begin, we consider $SO(8)$ gauge theories. An $SO(8)$ gauge theory can have $N_F$ fundamental, $N_S$ spinor, and $N_C$ conjugate spinor representations. When there is matter in these three representations, the Weyl chamber splits into six distinct subchambers. We study each of these chambers and find the marginal theories summarized in Table 9d. Due to the triality of the $D_4$ Dynkin diagram, given such an $SO(8)$ gauge theory, there exists another $SO(8)$ gauge theory obtained by permuting $N_F, N_S, N_C$. However, we stress that the physical Coulomb branches associated to different permutations are in general not identical. These results cover all known cases of $SO(8)$ gauge theories [26].

For $SO(9)$ gauge theories, there can be $N_F$ fundamental and $N_S$ spinors. When spinor matter is included, the fundamental Weyl chamber splits into three subchambers. We study each of these chambers and find the marginal theories summarized in Table 9e.

For $SU(5)$ gauge theories, we can consider hypermultiplets in the antisymmetric and the fundamental representations. We find four marignal theories as summarized in Table 9a. We would like to remark that our criteria for $SU(5)_{\frac{3}{2}} + 3\mathbf{AS} + 2\mathbf{F}$ is more involved. We find that this theory has a single sub-chamber which has a region $\mathcal{C}_{T>0}$ with positive string tensions and positive definite metric at infinite coupling. Therefore, naively, this theory is not marginal and it may have a 5d SCFT fixed point at UV. However, a careful analysis shows us that the boundary $\partial\mathcal{C}_{T>0}$ of the region is irrational. This indicates that this sub-chamber should not be a physical Coulomb branch. We expect that there exists a flop transition point where some non-perturbative instantons become massless so that we cannot reach the irrational boundary. We expect that, after taking into account the flop transition, this theory becomes marginal. The enhanced affine-type global symmetry of this theory in Table 9a supports this conjecture. The Coulomb branch with irrational boundaries and the geometric consideration associated to it will be discussed in Section 4.4.

Finally, we come to the case of $G = F_4$. Since the only representation with dimension smaller than the adjoint is the fundamental representation, an $F_4$ gauge theory can only have fundamental hypermultiplets. In this case, the Weyl chamber only consists of a single component, and therefore the task of determining a bound on the number $N_F$ of fundamental half-hypers is straightforward. We find that $N_F \leq 3$ and all these theories have 5d conformal fixed points at UV. So all the $F_4$ theories are non-marginal. The physical Coulomb branch of the theory with $N_F = 3$ is given by

$$\mathcal{C}_{\text{phys}} = \{2\phi_1 > \phi_2, \phi_1 + 2\phi_3 > 2\phi_2, 2\phi_4 \geq \phi_3, 2\phi_3 > \phi_2 + \phi_4\}. \tag{46}$$

**Rank 5,6,7,8**

For $SU(6)$ theories, we can have $N_{\text{TAS}}$ rank-3 antisymmetric hypermultiplets. Since **TAS** is pseudo-real, we can add half-hypermultiplets which we donote by $N_{\text{TAS}} = 1/2$. The marginal

Table 6: Marginal $SU(4)$ theories with CS level $\kappa$, $N_{\text{Sym}}$ symmetric, $N_{\text{AS}}$ antisymmetric, $N_{\text{F}}$ fundamental matters. The theories with negative CS level are obtained by $\phi_1 \leftrightarrow \phi_3$.

| $N_{\text{Sym}}$ | $N_{\text{AS}}$ | $N_{\text{F}}$ | $\kappa$ | $F$ | $\mathcal{C}_{\text{phys}}$ |
|---|---|---|---|---|---|
| 1 | 1 | 0 | 0 | $A_1^{(1)} \times U(1)$ | $2\phi_1 \geq \phi_2 \geq \phi_1 = \phi_3 \geq 0$ |
| 1 | 0 | 2 | 0 | $A_2^{(1)} \times U(1)$ | $\phi_1 = \phi_2 = \phi_3 \geq 0$ |
| 1 | 0 | 0 | 2 | $U(1)^2$ | $\phi_1 = \phi_2/2 = \phi_3/3 \geq 0$ |
| 0 | 4 | 0 | 4 | $E_6^{(2)}$ | $3\phi_1 \geq \phi_3 \geq \phi_1 = \phi_2/2 \geq 0$ |
| 0 | 4 | 0 | $1,2,3$ | $Sp(4) \times U(1)$ | $\phi_1 = \phi_2/2 = \phi_3 \geq 0$ |
| 0 | 4 | 0 | 0 | $A_7^{(2)}$ | $\phi_1 = \phi_2/2 = \phi_3 \geq 0$ |
| 0 | 3 | 4 | 0 | $A_{11}^{(2)} \times U(1)$ | $\phi_1 = \phi_2/2 = \phi_3 \geq 0$ |
| 0 | 3 | 4 | 1 | $Sp(7) \times U(1)$ | $\phi_1 = \phi_2/2 = \phi_3 \geq 0$ |
| 0 | 3 | 4 | 2 | $E_6^{(2)} \times SU(4)$ | $\phi_1 = \phi_2/2 = \phi_3 \geq 0$ |
| 0 | 3 | 0 | 5 | $Sp(3) \times U(1)$ | $\phi_1 = \phi_2/2 = \phi_3/3 \geq 0$ |
| 0 | 2 | 8 | 0 | $E_7^{(1)} \times B_3^{(1)}$ | $2\phi_1 \geq \phi_2 \geq \phi_1 = \phi_3 \geq 0$ |
| 0 | 2 | 7 | $\frac{3}{2}$ | $B_9^{(1)}$ | $\phi_1 = \phi_2/2 = \phi_3/2 \geq 0$ |
| 0 | 2 | 0 | 6 | $Sp(2) \times U(1)$ | $\phi_1 = \phi_2/2 = \phi_3/3 \geq 0$ |
| 0 | 1 | 10 | 0 | $A_{11}^{(1)}$ | $\phi_1 = \phi_2 = \phi_3 \geq 0$ |
| 0 | 1 | 8 | 2 | $E_8^{(1)} \times U(1)$ | $\phi_1 = \phi_2/2 = \phi_3/3 \geq 0$ |
| 0 | 1 | 0 | 7 | $SU(2) \times U(1)$ | $\phi_1 = \phi_2/2 = \phi_3/3 \geq 0$ |
| 0 | 0 | 12 | 0 | $D_{12}^{(1)}$ | $\phi_1 = \phi_2 = \phi_3 \geq 0$ |
| 0 | 0 | 8 | 3 | $E_8 \times U(1)$ | $\phi_1 = \phi_2/2 = \phi_3/3 \geq 0$ |
| 0 | 0 | 0 | 8 | $U(1)$ | $\phi_1 = \phi_2/2 = \phi_3/3 \geq 0$ |

$SU(6)$ theories are summarized in Table 10a. For $SU(7)$ theories, we can also have rank-3 antisymmetric hypermultiplets, but only a single full-hypermultiplet is allowed. All the exceptional cases have $N_{\text{TAS}} = 1$. The $SU(7)$ theories are summarized in Table 11a. The first two of them are marginal. However, due to computational limits, we could not clearly identify the last theory whether it is non-marginal or exotic, which will be discussed in Section 4.4. For $SU(8)_\kappa + N_{\text{TAS}}\text{TAS}$ we find that there are no non-trivial exceptional $SU(8)$ theories for any value of $N_{\text{TAS}}$.

For $SO(10)$ theories, we can have $N_{\text{S}}$ spinor and $N_{\text{F}}$ fundamental hypermultiplets. The marginal $SO(10)$ theories are summarized in Table 10b. For $SO(11)$ theories, we can have $2N_{\text{S}}$ spinor half-hypermultiplets and $N_{\text{F}}$ vector hypermultiplets. The marginal $SO(11)$ theories are summarized in Table 10c. For $SO(12)$ theories, we can have $N_{\text{F}}$ fundamental hypermultiplets, along with $2N_{\text{S}}$ spinor and $2N_{\text{C}}$ conjugate spinor half-hypermultiplets. So we have a large number of exceptional theories as summarized in Table 11b. Note that the last theory in this table could either be non-marginal or exotic, as we are presently unable to verify its status due to computational limitations. All the other theories are marginal. For $SO(13)$ theories,

Table 7: Marginal $Sp(3)$ gauge theories with $N_{\text{TAS}}$ rank-3 antisymmetric, $N_{\text{AS}}$ anti-symmetric, $N_{\text{F}}$ fundamental matters.

| $N_{\text{TAS}}$ | $N_{\text{AS}}$ | $N_{\text{F}}$ | $F$ | $\mathcal{C}_{\text{phys}}$ |
|---|---|---|---|---|
| 1 | 0 | 5 | $E_6^{(1)}$ | $3\phi_1 \geq \phi_3 \geq 2\phi_1 = \phi_2 \geq 0$ |
| $\frac{1}{2}$ | 1 | $\frac{5}{2}$ | $Sp(4)$ | $\phi_1 = \phi_2/2 = \phi_3/3 \geq 0$ |
| $\frac{1}{2}$ | 0 | $\frac{19}{2}$ | $SO(19) \times U(1)$ | $2\phi_1 \geq \phi_2 = \phi_3 \geq \phi_1 \geq 0$ |
| 0 | 2 | 0 | $C_2^{(1)}$ at $\theta = 0$, $Sp(2) \times U(1)$ at $\theta = \pi$ | $\phi_1 = \phi_2/2 = \phi_3/3 \geq 0$ |
| 0 | 1 | 8 | $E_8^{(1)} \times SU(2)$ | $\{2\phi_1 \geq \phi_3 \geq \phi_1 \cap \phi_3 \geq \phi_2 \geq (\phi_1+\phi_3)/2\} \cup$ $\{3\phi_1 \geq \phi_3 \geq 2\phi_1 \cap 2\phi_1 \geq \phi_2 \geq (\phi_1+\phi_3)/2\}, \phi_1 \geq 0$ |
| 0 | 0 | 12 | $D_{12}^{(1)}$ | $\phi_1 = \phi_2 = \phi_3 \geq 0$ |

Table 8: Marginal $SO(7)$ gauge theories with $N_{\text{S}}$ spinor and $N_{\text{F}}$ fundamental matters.

| $N_{\text{S}}$ | $N_{\text{F}}$ | $F$ | $\mathcal{C}_{\text{phys}}$ |
|---|---|---|---|
| 7 | 0 | $Sp(7) \times U(1)$ | $\phi_1 = \phi_2/2 = \phi_3 \geq 0$ |
| 6 | 1 | $A_{11}^{(2)} \times SU(2)$ | $\phi_1 = \phi_2/2 = \phi_3 \geq 0$ |
| 5 | 2 | $C_7^{(1)}$ | $\phi_1 = \phi_2/2 = \phi_3 \geq 0$ |
| 4 | 3 | $E_6^{(2)} \times A_5^{(2)}$ | $\phi_1 = \phi_2 \geq \phi_3 \geq \phi_1/2 \geq 0$ |
| 2 | 4 | $A_{12}^{(2)}$ | $\phi_1 = \phi_2 = 2\phi_3 \geq 0$ |

we can have $2N_{\text{S}}$ spinor half-hypermultiplets and the marginal theories are summarized in Table 11c. $SO(14)$ case has merely one exceptional theory with a $N_{\text{S}} = 1$ spinor (or $N_{\text{C}} = 1$ conjugate spinor) and $N_{\text{F}} = 6$ fundamental hypermultiplets as listed in Table 11d. An $E_6$ gauge theory can have $N_{\text{F}} \leq 4$ fundamental hypermultiplets and an $E_7$ gauge theory can have $2N_{\text{F}} \leq 6$ fundamental half-hypermultilets. However, we are not sure whether or not the theories $E_6 + 4\mathbf{F}$ and $E_7 + 3\mathbf{F}$ are exotic. See Section 4.4 for more discussions about this. The $\mathcal{N} = 1$ $E_8$ gauge theory cannot have any hypermultiplets, and the $E_8$ theory without matter flows to a 5d SCFT at the UV fixed point.

## 4.4 Exotic exceptional theories

In our main conjecture, we require that a physical Coulomb branch should have rational boundary. In geometry this means that a physical Coulomb branch must be determined as a subregion of the Weyl chamber bounded by hyperplanes where some of 4-cycles shrink to 2-cycles or contract to a point. In addition, it is necessary that when a 4-cycle shrinks to a 2-cycle, there must exist an associated shrinking 2-cycle. Such hyperplanes can be characterized by the condition that some of string tensions vanish, i.e. $T_i(\phi) = 0$ for some $i$. The rational boundary condition can therefore be recast as $\sum_j n_j \phi_j = 0$ with $n_j \in \mathbb{Z}$ for each hyperplane.

However, we notice that there are some cases where sub-chambers are bounded by irrational boundaries at $\sum_j n_j \phi_j = 0$ with $n_j \in \mathbb{R} \backslash \mathbb{Q}$. Such irrational boundaries are not allowed

Table 9: Rank 4 exceptional gauge theories.

(a) Exceptional $SU(5)_\kappa$ gauge theories with $N_{AS}$ antisymmetric, $N_F$ fundamental matters. The first three are marginal theoriese and the last one is an exotic theory.

| $N_{AS}$ | $N_F$ | $|\kappa|$ | $F$ |
|---|---|---|---|
| 3 | 3 | 0 | $C_3^{(1)} \times SU(3)$ |
| 3 | 1 | 3 | $SU(4) \times SU(2) \times U(1)$ |
| 0 | 5 | $\frac{11}{2}$ | $SO(10) \times U(1)$ |
| 3 | 2 | $\frac{3}{2}$ | $E_6^{(2)} \times U(1)$ |

(b) Marginal $Sp(4)$ gauge theories with $N_{TAS}$ rank-3 antisymmetric, $N_F$ fundamental matters.

| $N_{TAS}$ | $N_F$ | $F$ |
|---|---|---|
| $\frac{1}{2}$ | 4 | $SO(8) \times U(1)$ |

(c) Exceptional $F_4$ gauge theories with $N_F$ fundamental matters.

| $N_F$ | $F$ |
|---|---|
| 3 | $Sp(3) \times SU(2)$ |

(d) Marginal $SO(8)$ gauge theories with $N_S$ spinor, $N_C$ conjugate spinor, $N_F$ fundamental matters. Due to triality of the $D_4$ Dynkin diagram, for each of the above theories there exists an entire family of theories obtained by permutation of the values $(N_F, N_S, N_C)$.

| $N_S$ | $N_C$ | $N_F$ | $F$ |
|---|---|---|---|
| 5 | 2 | 0 | $A_{14}^{(2)}$ |
| 5 | 1 | 1 | $A_{13}^{(2)}$ |
| 4 | 4 | 0 | $E_6^{(2)} \times E_6^{(2)}$ |
| 4 | 3 | 1 | $F_4^{(1)} \times Sp(4)$ |
| 4 | 2 | 2 | $A_7^{(2)} \times D_7^{(2)}$ |
| 3 | 3 | 2 | $C_6^{(1)} \times Sp(2)$ |

(e) Marginal $SO(9)$ gauge theories with $N_S$ spinor, $N_F$ fundamental matters.

| $N_S$ | $N_F$ | $F$ |
|---|---|---|
| 4 | 1 | $E_6^{(2)} \times SU(2)$ |
| 3 | 3 | $C_3^{(1)} \times Sp(3)$ |
| 2 | 5 | $A_9^{(2)} \times A_4^{(2)}$ |
| 1 | 6 | $A_{14}^{(2)}$ |

due to geometric considerations, as the class of the would-be 2-cycle that vanishes on such a boundary would have irrational coefficients and therefore be ill-defined. We also remark that the dictionary between geometry and field theory teaches us that these boundaries can equivalently be characterized by the vanishing of some BPS particle with irrational central charge, and therefore we find we must also dismiss such boundaries on physical grounds. This may imply that either we cannot enter such sub-chambers or that we will encounter a geometric transition such as a flop transition before we meet the irrational hyperplane, which cannot be captured by our perturbative analysis. When this happens, we admit that our perturbative analysis using the condition $T_i(\phi) = 0$ cannot determine the physical Coulomb branch in the sub-chamber exactly. If a theory has another sub-chamber with rational boundaries, we can apply our criteria for the sub-chamber and classify the theory accordingly. However, when all the sub-chambers are bounded by irrational hyperplanes, it turns out to be rather difficult to determine whether or not the theory has a 5d CFT fixed point. We find a handful of such *exotic* theories, which we describe in more detail below.

For example, $5\mathbf{F} + SU(2) \times SU(2) + 2\mathbf{F}$ has a single physical sub-chamber with irrational boundaries. The sub-chamber is given by

$$\frac{2}{3}\phi_1 < \phi_2 < \frac{1}{\sqrt{2}}\phi_1 \,, \tag{47}$$

where $\phi_1, \phi_2$ are scalar vevs of two $SU(2)$ gauge groups respectively. In the above sub-chamber, string tensions are positive and moreover the metric is positive definite at infinite

Table 10: Rank 5 exceptional gauge theories.

(a) Marginal $SU(6)_\kappa$ gauge theories with $N_{\text{TAS}}$ rank-3 antisymmetric, $N_{\text{Sym}}$ symmetric, $N_{\text{AS}}$ antisymmetric, $N_{\text{F}}$ fundamental matters.

| $N_{\text{TAS}}$ | $N_{\text{Sym}}$ | $N_{\text{AS}}$ | $N_{\text{F}}$ | $|\kappa|$ | $F$ |
|---|---|---|---|---|---|
| 2 | 0 | 0 | 0 | 0 | $A_2^{(2)} \times A_2^{(2)}$ |
| $\frac{3}{2}$ | 0 | 0 | 5 | 0 | $SO(3){\times}SU(5){\times}U(1)^2$ |
| $\frac{3}{2}$ | 0 | 0 | 3 | 2 | $SO(3){\times}SU(3){\times}U(1)^2$ |
| $\frac{3}{2}$ | 0 | 0 | 0 | $\frac{9}{2}$ | $SO(3) \times U(1)^2$ |
| 1 | 0 | 1 | 4 | 0 | $D_4^{(1)} \times A_1^{(1)} \times U(1)$ |
| 1 | 0 | 1 | 3 | $\frac{3}{2}$ | $A_2^{(2)} \times A_2^{(2)} \times SU(4)$ |
| 1 | 0 | 1 | 0 | 4 | $SU(2) \times U(1)^2$ |
| 1 | 0 | 0 | 10 | 0 | $D_{10}^{(1)}$ |
| 1 | 0 | 0 | 9 | $\frac{3}{2}$ | $E_8^{(1)} \times A_2^{(1)}$ |
| $\frac{1}{2}$ | 1 | 0 | 1 | 0 | ? |
| $\frac{1}{2}$ | 1 | 0 | 0 | $\frac{3}{2}$ | ? |
| $\frac{1}{2}$ | 0 | 2 | 2 | $\frac{3}{2}$ | $D_4^{(3)} \times SU(2) \times U(1)$ |
| $\frac{1}{2}$ | 0 | 2 | 2 | $\frac{1}{2}$ | $SU(5) \times U(1)$ |
| $\frac{1}{2}$ | 0 | 2 | 0 | $\frac{7}{2}$ | $SU(3) \times U(1)$ |
| $\frac{1}{2}$ | 0 | 1 | 9 | 0 | $SU(11) \times U(1)$ |
| $\frac{1}{2}$ | 0 | 1 | 8 | $\frac{3}{2}$ | $SO(16){\times}SU(2){\times}U(1)$ |
| $\frac{1}{2}$ | 0 | 0 | 13 | 0 | $SU(13) \times U(1)^2$ |
| $\frac{1}{2}$ | 0 | 0 | 9 | 3 | $SU(9) \times U(1)^2$ |
| 0 | 0 | 3 | 0 | 3 | $D_4^{(3)} \times SU(2)$ |
| 0 | 0 | 3 | 0 | 1 | $Sp(3) \times U(1)$ |
| 0 | 0 | 3 | 0 | 0, 2 | $SU(3) \times U(1)^2$ |
| 0 | 0 | 0 | 0 | 9 | $U(1)$ |

(b) Marginal $SO(10)$ gauge theories with $N_{\text{S}}$ spinor, $N_{\text{C}}$ conjugate spinor, $N_{\text{F}}$ fundamental matters.

| $N_{\text{S}}$ | $N_{\text{F}}$ | $F$ |
|---|---|---|
| 4 | 2 | $B_3^{(1)} \times SU(4)$ |
| 3 | 4 | $C_3^{(1)} \times Sp(4)$ |
| 2 | 6 | $A_{11}^{(2)} {\times} A_2^{(2)} {\times} A_1^{(1)}$ |
| 1 | 7 | $A_{15}^{(2)}$ |

(c) Marginal $SO(11)$ gauge theories with $N_{\text{S}}$ spinor, $N_{\text{F}}$ fundamental matters. We use $U(1)^{(2)}$ for a $U(1)$ compactified with an outer automorphism twist.

| $N_{\text{S}}$ | $N_{\text{F}}$ | $F$ |
|---|---|---|
| $\frac{5}{2}$ | 0 | $Sp(2) \times U(1)$ |
| 2 | 3 | $A_5^{(2)} {\times} A_2^{(2)} {\times} A_2^{(2)}$ |
| $\frac{3}{2}$ | 5 | $A_2^{(2)} \times Sp(5) \times U(1)^{(2)}$ |
| 1 | 7 | $A_{13}^{(2)} \times A_1^{(1)} \times U(1)^{(2)}$ |
| $\frac{1}{2}$ | 8 | $Sp(9)$ |

coupling. However, it is clear from the above expression that one of the boundaries is defined by an irrational linear combination of Coulomb branch parameters,

$$\phi_1 + \sqrt{2}\phi_2 = 0. \tag{48}$$

From geometric considerations, we believe this boundary cannot correspond to the vanishing of a 4-cycle with rational divisor class, and hence the correspondence between BPS data and geometric data is invalid in this description. This is an indication that a flop transition of some sort—possibly due to the appearance of massless instantons—must occur before we meet this wall on the Coulomb branch. The presence of spurious boundaries of the above type is indicative of our inability to compute instanton masses and other non-perturbative data using only the perturbative expression for the prepotential.

Table 11: Rank 6,7,8 exceptional gauge theories.

(a) Exceptional $SU(7)_\kappa$ gauge theories with $N_{\text{TAS}}$ rank-3 antisymmetric and $N_{\text{F}}$ fundamental matters. The first two are marginal, while the last one could be non-marginal or exotic.

| $N_{\text{TAS}}$ | $N_{\text{F}}$ | $|\kappa|$ | $F$ |
|---|---|---|---|
| 1 | 6 | 0 | $D_6^{(1)}$ |
| 1 | 5 | $\frac{3}{2}$ | $SO(10) \times SU(2) \times U(1)$ |
| 1 | 0 | 5 | $U(1)^2$ |

(b) Exceptional $SO(12)$ gauge theories with $N_{\text{S}}$ spinor, $N_{\text{C}}$ conjugate spinor, $N_{\text{F}}$ fundamental matters. The last theory could be non-marginal or exotic and the other theories are all marginal.

| $N_{\text{S}}$ | $N_{\text{C}}$ | $N_{\text{F}}$ | $F$ |
|---|---|---|---|
| $\frac{5}{2}$ | 0 | 0 | $Sp(2) \times U(1)$ |
| 2 | 0 | 4 | $E_6^{(2)} \times A_2^{(2)} \times A_2^{(2)}$ |
| $\frac{3}{2}$ | 1 | 1 | $SU(4) \times SU(2)$ |
| $\frac{3}{2}$ | $\frac{1}{2}$ | 4 | $A_2^{(2)} \times Sp(4)$ |
| $\frac{3}{2}$ | 0 | 6 | $Sp(6) \times SU(2) \times U(1)$ |
| 1 | 1 | 4 | $A_7^{(2)} \times A_1^{(1)} \times A_1^{(1)} \times U(1)^{(2)}$ |
| 1 | $\frac{1}{2}$ | 6 | $Sp(6) \times SU(2)^2$ |
| 1 | 0 | 8 | $A_{15}^{(2)} \times A_1^{(1)}$ |
| $\frac{1}{2}$ | $\frac{1}{2}$ | 8 | $A_{15}^{(2)} \times U(1)^{(2)}$ |
| $\frac{1}{2}$ | 0 | 9 | $Sp(9) \times U(1)$ |
| 2 | $\frac{1}{2}$ | 0 | $SU(4)$ |

(c) Marginal $SO(13)$ gauge theories with $N_{\text{S}}$ spinor, $N_{\text{F}}$ fundamental matters.

| $N_{\text{S}}$ | $N_{\text{F}}$ | $F$ |
|---|---|---|
| 1 | 5 | $A_9^{(2)} \times A_1^{(1)}$ |
| $\frac{1}{2}$ | 9 | $A_{17}^{(2)}$ |

(d) Marginal $SO(14)$ gauge theories with $N_s$ spinor, $N_{\text{F}}$ fundamental matters.

| $N_{\text{S}}$ | $N_{\text{F}}$ | $F$ |
|---|---|---|
| 1 | 6 | $A_{11}^{(2)} \times A_1^{(1)}$ |

(e) Exceptional $E_6$ gauge theories with $N_{\text{F}}$ fundamental matters. This theory can be non-marginal or exotic.

| $N_{\text{F}}$ | $F$ |
|---|---|
| 4 | $SU(4) \times SU(2) \times U(1)$ |

(f) Exceptional $E_7$ gauge theories with $N_{\text{F}}$ fundamental matters. This theory can be non-marginal or exotic.

| $N_{\text{F}}$ | $F$ |
|---|---|
| 3 | ? |

(g) $E_8$ gauge theory has no matters.

| $N_{\text{F}}$ | $F$ |
|---|---|
| 0 | $\emptyset$ |

It turns out that the $5\mathbf{F} + SU(2) \times SU(2) + 2\mathbf{F}$ theory is dual to $SU(3)_{\frac{3}{2}} + 9\mathbf{F}$. These two dual theories admit the same brane construction in Type IIB string theory. As we discussed in Section 4.3, the dual $SU(3)$ gauge theory has non-trivial 6d completion in the UV fixed point having a zero eigenvalue of the metric at infinite coupling. In fact, we can obtain the $SU(2) \times SU(2)$ gauge theory description from the $SU(3)$ gauge theory by turning on mass parameters such as $a_1 > m_{1,2,3,4,5,6,7} > a_2$, $a_2 > m_{8,9} > a_3$ where $a_{1,2,3}$ are Coulomb parameters of $SU(3)$ and $m_i$ is the mass of the $i$-th hypermultiplet. This corresponds to two consecutive flop transitions in the corresponding geometry. This supports our expectation that the geometry undergoes flop transitions before we meet the irrational boundary in the Coulomb branch. This example also shows that while we may not be able to reach the UV fixed point with the $SU(2) \times SU(2)$ gauge theory description, the 6d fixed point can nevertheless be attained by way of the $SU(3)$ gauge theory description after the flop transitions. Therefore, when we have only physical Coulomb branches bounded by irrational curves, then the naive application of our criteria cannot tell us whether or not the theory is marginal. We call such theory as *exotic*.

In our classification, we have encountered five possible cases of exotic theories, only one of which we have explicitly confirmed as such. They are:

1. $SU(5)_{\frac{3}{2}} + 3\mathbf{AS} + 2\mathbf{F}$  (confirmed)

2. $SU(7)_5 + 1\mathbf{TAS}$  (undetermined)

3. $SO(12) + 2\mathbf{S} + \frac{1}{2}\mathbf{C}$  (undetermined)

4. $E_6 + 4\mathbf{F}$  (undetermined)

5. $E_7 + 3\mathbf{F}$  (undetermined)

For the first case above, the $SU(5)$ theory, $\mathcal{C}_{\text{phys}}$ consists of a single component with irrational boundaries. Within this chamber, as expected, the string tensions are positive and the metric is positive definite. However, we expect that this theory has a 6d fixed point based on the affinization of the flavor symmetry $E_6^{(2)} \times U(1)$ according to a 1-instanton analysis. However, if this is case, it remains unclear what dual physical description we must introduce (as in the case of $5\mathbf{F} + SU(2) \times SU(2) + 2\mathbf{F}$) in order to reach the 6d fixed point through the Coulomb branch.

We have also numerically confirmed the existence of physical Coulomb branches $\mathcal{C}_{\text{phys}}$ for the other theories in the above list. Unfortunately, due to computational limitations, we have been unable to precisely test whether or not the boundaries of these regions are rational or irrational. For this reason, it is unclear whether or not these theories are exotic in the sense we have described above. However, we note the two exceptional theories in the above list each have two distinct candidates for their 6d lifts, namely 6d SCFTs corresponding to a single curve with self-intersection $-n$, where $n = 1, 2$. In the case of $E_6$, the $n = 1$ theory has $N_{\mathbf{F}} = 5$, while the $n = 2$ theory has $N_{\mathbf{F}} = 4$. In the case of $E_7$, the $n = 1$ theory has $N_{\mathbf{F}} = 7/2$, while the $n = 2$ theory has $N_{\mathbf{F}} = 3$.

# 5   6d Theories on a Circle

In this section we present some 6d SCFTs that are the lifts of some of the marginal 5d theories we found. We note that most of these theories have already been discussed elsewhere in the literature. We discuss the new theories in detail, while for the known theories we merely quote the results.

## 5.1   Generic $Sp(N)$ and $SU(N)$ with fundamental and antisymmetric matter

We start with the cases $Sp(N)$ and $SU(N)$, for generic $N$, with matter in the fundamental and antisymmetric representations. These theories can be constructed with ordinary brane web systems, where the $Sp(N)$ and $SU(N)$ theories with antisymmetric matter can be brought to this state by using an $O7^-$-plane and then decomposing it into a pair of 7-branes [17]. The 6d lifts of these types of theories were studied extensively in [14, 21, 31, 33], and we shall quote their results for the theories of interest.

The 6d SCFTs appearing in these cases can all be engineered by a brane system involving NS5-branes, D6-branes and D8-branes in the presence of an $O8^-$-plane. When compactified and T-dualized the system becomes a brane configuration of NS5-branes, D5-branes and D7-branes in the presence of two $O7^-$-planes. Upon decomposing the two $O7^-$-planes to a pair of 7-branes, one obtains the brane webs of the associated 5d gauge theories.

## $SU(N)_0 + (2N+4)\textbf{F}$ and $Sp(N-1) + (2N+4)\textbf{F}$

The 5d gauge theory $SU(N)_0 + (2N+4)\textbf{F}$ is known to lift to 6d [34]. The 6d lift was worked out in [15, 31], and is the so-called $(D_{N+4}, D_{N+4})$ conformal matter [35]. This theory has a low-energy description on the tensor branch as the 6d gauge theory $Sp(N-2) + (2N+4)\textbf{F}$. In F-theoretic notation, this theory is denoted

$$\overset{\mathfrak{sp}_{N-2}}{1} \quad [SO(4N+8)].$$

It is straightforward to see that the dimensions of the Coulomb branch in both descriptions agree. Also the 1-instanton spectrum of the 5d theory is consistent with an enhanced $D_{2N+4}^{(1)}$, agreeing with the spectrum expected from the compactification of the 6d theory on a circle.

The 5d gauge theory $Sp(N-1) + (2N+4)\textbf{F}$ also lifts to 6d and is in fact dual to $SU(N)_0 + (2N+4)\textbf{F}$ on account of both theories lifting to the same 6d SCFT [31].

## $SU(N)_0 + 1\textbf{AS} + (N+6)\textbf{F}$

The 5d gauge theory $SU(N)_0 + 1\textbf{AS} + (N+6)\textbf{F}$ lifts to the 6d SCFT which has a low-energy description on the tensor branch as the 6d gauge theory $SU(N-1) + 1\textbf{AS} + (N+7)\textbf{F}$ [21, 31]. In F-theoretic notation this 6d SCFT is denoted

$$\overset{\mathfrak{su}_{N-1}}{1} \quad [SU(N+7)].$$

Again it is easy to see that the Coulomb branch dimensions agree. The 6d and 5d theories both have a Higgs branch, associated with giving a vev to the antisymmetric matter, upon which we reduce to the previous case.

We can also compare the global symmetries of these theories, which are summarized in Tables 1a and 2. For $N > 4$ The 1-instanton analysis suggests that the current spectrum of the 5d theory is consistent with $A_{N+6}^{(1)}$ which again agrees with the 6d theory on a circle. There are however additional currents when $N \leq 5$. For $N = 3$ an antisymmetric hypermultiplet is identical to an antifundamental hypermultiplet so in 5d this case is identical to the previous case. This feature is correctly captured by the 6d description, as for $SU(2)$ the antisymmetric representation is a singlet. For $N = 4$ the antisymmetric represenation is real and the $U(1)$ associated with this representation in the general case is perturbatively enhanced to $SU(2)$. There are then additional 1-instanton currents, and the spectrum of the 5d theory is consistent with a larger enhancement to $A_{11}^{(1)}$. This again agrees with the perturbative enhancement in 6d.

Finally when $N = 5$ we also get additional currents, and the spectrum is consistent with an enhancement to $A_{11}^{(1)} \times A_1^{(1)}$. This again agrees with the perturbative enhancement in 6d.

## $SU(N)_0 + 2\textbf{AS} + 8\textbf{F}$

The 5d gauge theory $SU(N)_0 + 2\textbf{AS} + 8\textbf{F}$ has different 6d lifts depending on whether $N$ is even or odd [21]. The 6d lift can be described on the tensor branch as a low-energy 6d semi-gauge theory. The description for these two cases is shown in Figure 9, where (a) is the $N = 2n$ case, and (b) is the $N = 2n+1$ case. Alternatively, we also provide an F-theoretic description of both cases,

$$[E_7] \quad 1 \quad \overset{\mathfrak{su}_2}{2} \quad \overset{\mathfrak{su}_2}{2} \quad \cdots \quad \overset{\mathfrak{su}_2}{2} \quad \overset{\mathfrak{su}_2}{2} \quad [SU(2)]$$
$$[SU(2)],$$

$$[SO(16)] \quad \overset{\mathfrak{sp}_1}{1} \quad \overset{\mathfrak{su}_2}{2} \quad \overset{\mathfrak{su}_2}{2} \quad \cdots \quad \overset{\mathfrak{su}_2}{2} \quad \overset{\mathfrak{su}_2}{2} \quad [SU(2)],$$

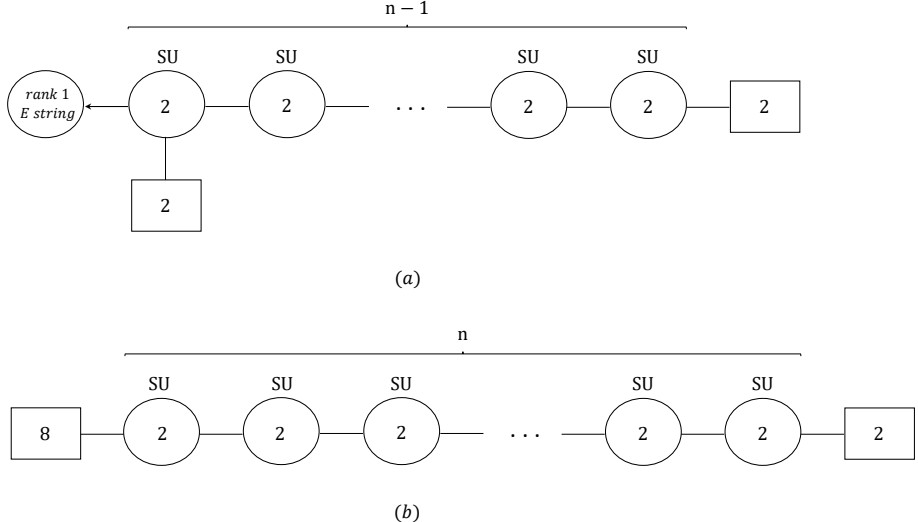

Figure 9: The low-energy description on the tensor branch of the 6d SCFTs which are the lifts of the 5d $SU(N)_0 + 2\mathbf{AS} + 8\mathbf{F}$ gauge theories. Quiver (a) describes the case $N = 2n$, while quiver (b) the case $N = 2n + 1$. In quiver (a) the arrow into the rank 1 E-string theory represents gauging a subgroup $SU(2) \subset E_8$ of the global symmetry group of the E-string theory.

where the upper diagram is the $N = 2n$ case, the lower diagram is the $N = 2n + 1$ case. Note that there are $n - 1$ **2** curves in both diagrams above.

One can see that the Coulomb branch dimensions of the 5d theories and 6d theories on a circle agree. The global symmetries also agree, where instantons in the 5d gauge theory lead to additional conserved currents consistent with what is expected from the 6d theory on a circle. The results for the various cases are summarized in Tables 1a and 2.

## $SU(N)_{\pm\frac{3}{2}} + 2\mathbf{AS} + 7\mathbf{F}$

The 5d gauge theory $SU(N)_{\pm\frac{3}{2}} + 2\mathbf{AS} + 7\mathbf{F}$ is also known to lift to 6d, and again goes to different 6d theories depending on whether $N$ is even or odd. The cases $N = 3, 4, 5$ were explicitly discussed in [21] and it is straightforward to generalize the construction to generic $N$. The 6d SCFTs can be again conveniently represented using the low-energy description on the tensor branch. Descriptions of the two cases are shown in Figure 9, where (a) is the $N = 2n$ case, and (b) is the $N = 2n + 1$ case. We also provide F-theoretic descriptions,

$$[SO(16)] \quad \overset{\mathfrak{sp}_1}{1} \quad \overset{\mathfrak{su}_2}{2} \quad \overset{\mathfrak{su}_2}{2} \quad \cdots \quad \overset{\mathfrak{su}_2}{2} \quad \underset{[SU(2)]}{\overset{\mathfrak{su}_2}{2}} \quad \overset{II}{2} \quad ,$$

$$[E_7] \quad 1 \quad \underset{[SU(2)]}{\overset{\mathfrak{su}_2}{2}} \quad \overset{\mathfrak{su}_2}{2} \quad \cdots \quad \overset{\mathfrak{su}_2}{2} \quad \underset{[SU(2)]}{\overset{\mathfrak{su}_2}{2}} \quad \overset{II}{2} \quad ,$$

where the upper diagram is the $N = 2n$ case and the lower diagram is the $N = 2n + 1$ case. Note that there are $n - 1$ **2** curves in the upper diagram and $n$ **2** curves in the lower diagram.

One can see that the Coulomb branch dimensions of the 5d theories and 6d theories on a circle agree. The global symmetries also appear to agree, at least as far as can be seen from the 1-instanton analysis. Again the results for the various cases are summarized in Tables 1a and 2.

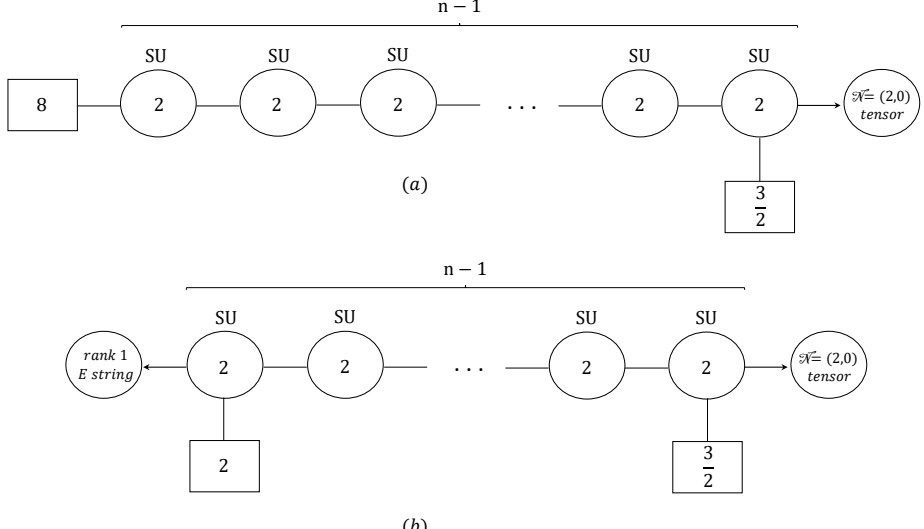

Figure 10: The low-energy description on the tensor branch of the 6d SCFTs which are the lifts of the 5d $SU(N)_{\pm\frac{3}{2}} + 2\mathbf{AS} + 7\mathbf{F}$ gauge theory. Quiver (a) describes the case $N = 2n$ while quiver (b) the case $N = 2n + 1$. The arrow into the $\mathcal{N} = (2, 0)$ tensor, which appears on the rightmost in both quivers, represents gauging the $SU(2)$ global symmetry of the tensor (this is part of its $Sp(2)$ R-symmetry that is seen as a global symmetry from the $\mathcal{N} = (1, 0)$ point of view).

### $Sp(N-1) + 1\mathbf{AS} + 8\mathbf{F}$ and $SU(N)_{\pm\frac{N}{2}} + 1\mathbf{AS} + 8\mathbf{F}$

The case of $Sp(N-1) + 1\mathbf{AS} + 8\mathbf{F}$ is well known to lift to the rank $N-1$ E-string theory [36]. On the other hand, to our knowledge $SU(N)_{\pm\frac{N}{2}} + 1\mathbf{AS} + 8\mathbf{F}$, for $N > 3$, has not been discussed in the literature. From brane webs we can argue that the 5d gauge theory $SU(N)_{\pm\frac{N}{2}} + 1\mathbf{AS} + 8\mathbf{F}$ should have a 6d lift.

To argue this we consider the brane web description of this theory, which can be engineered using an $O7^-$-plane and resolving it. The brane web looks slightly different depending on whether $N$ is even or odd. For simplicity, we shall show here only the $N$ even case, though most of the steps and results apply also to the $N$ odd case.

Figure 11 shows the web for an $SU(N)_{\frac{N-N_\mathbf{F}+8}{2}} + 1\mathbf{AS} + N_\mathbf{F}\mathbf{F}$ gauge theory. For simplicity we have only shown the external legs and not how these connect inside the web, as that is irrelevant for this discussion. By manipulating the web we arrive at the web on the bottom left of Figure 11. When $N > 10$ one of the legs collides with another leg and further transitions ensue. We can calculate the monodromy "felt" by this leg when going around the entire web, and we find it to be the $SL(2, \mathbb{Z})$ transformation $T^{8-N_\mathbf{F}}$. Thus we see that if $N_\mathbf{F} < 8$ and the transition does not terminate before going full circle, the brane will come back less divergent. Therefore this process will terminate after a finite number of rotations. In particular, for $N_\mathbf{F} = 0$, this means that a fixed point exists for every $N$. For $N_\mathbf{F} = 8$ the brane returns back to itself while for $N_\mathbf{F} > 8$ the brane returns more divergent.

This suggests then that for $N_\mathbf{F} < 8$ a 5d fixed point exists, while for $N_\mathbf{F} > 8$ neither a 5d nor a 6d fixed point exists. The case $N_\mathbf{F} = 8$ seems to be a 6d lift. For $N \leq 10$, the process already terminates at the step shown in the bottom left of Figure 11, and we can show this explicitly. Furthermore we can argue that this theory is dual to $Sp(N-1) + 1\mathbf{AS} + N_\mathbf{F}\mathbf{F}$.

This follows from 7-brane manipulations. We have shown the relevant steps in Figure 12 for the simpler case of $N = 6$. The manipulations for the other cases are similar, and we omit

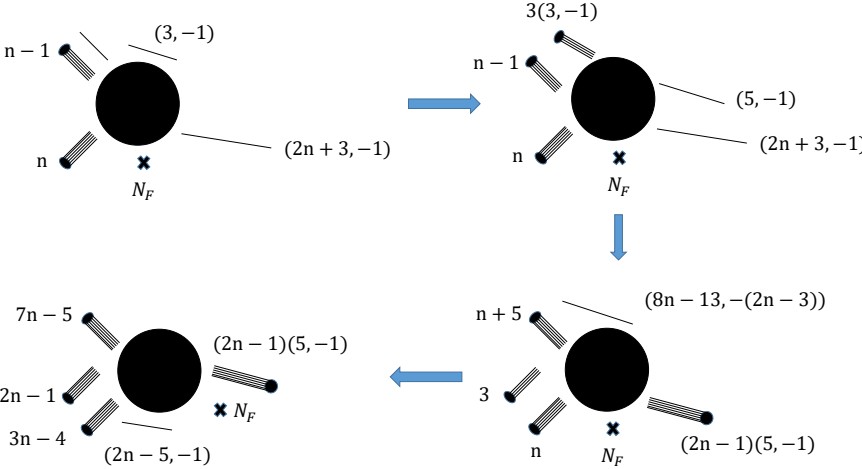

Figure 11: Brane web manipulation for $SU(N)_{\frac{N-N_{\mathbf{F}}+8}{2}} + 1\mathbf{AS} + N_{\mathbf{F}}\mathbf{F}$, in the case $N = 2n$. For ease of drawing, only the external legs are shown. We use a black '×' to represent the $N_{\mathbf{F}}$ 7-branes that give the flavors. The picture on the top left is for the initial web. Moving the $(3, -1)$ 7-brane through the single $(1, -1)$ 5-brane leads to the picture in the top right. Further moving the $(3, -1)$ 7-brane through the $n - 1$ $(1, -1)$ 5-branes leads to the picture in the bottom right, where we have also moved the $(5, -1)$ 7-brane through the $(2n + 3, -1)$ 5-brane. Finally, we move the $(8n - 13, -(2n - 3))$ 7-brane through all the $(1, -1)$ and $(1, 1)$ 5-branes leading to the picture at the bottom left. If $n \leq 5$ then no further transitions are required.

them for brevity. Therefore, in that range at least, the $SU(N)_{\frac{N-N_{\mathbf{F}}+8}{2}} + 1\mathbf{AS} + N_{\mathbf{F}}\mathbf{F}$ class of theories go to a 5d fixed point for $N_{\mathbf{F}} < 8$, and the $SU(N)_{\pm\frac{N}{2}} + 1\mathbf{AS} + 8\mathbf{F}$ theory should lift to the 6d rank $N - 1$ E-string theory. We suspect this to be true for all $N$.

## 5.2 Generic $SO(N)$ and $SU(N)$ with symmetric matter

Next we examine the cases of $SO(N)$ and $SU(N)$ theories with matter in the symmetric representation for generic $N$. This class of theories can be engineered in string theory by brane webs in the presence of O7$^+$-planes [17]. The 6d lifts for $SO(N)$ with fundamental matter, and $SU(N)$ with symmetric and fundamental matter were studied in [22], and it is straightforward to extend these results to some of the other cases.

The 6d SCFTs in these cases can be engineered in string theory by a brane system involving NS5-branes, D6-branes and D8-branes, but the compactification is done with a $\mathbb{Z}_2$ twist in a discrete symmetry. In the brane system this discrete symmetry is manifested as worldsheet parity combined with a reflection in the direction orthogonal to the D8-branes. Performing T-duality, the authors of [22] mapped this system to a configuration involving NS5-branes and D5-branes in the presence of an O7$^+$ and O7$^-$-planes. Decomposing the O7$^-$-plane then leads to the brane webs for the 5d gauge theories.

### $SO(N) + (N-2)\mathbf{F}$

The case of 5d $SO(N) + (N-2)\mathbf{F}$ gauge theory is known to lift to a 6d SCFT. The 6d SCFT was worked out in [22] to be the one living on 2 M5-branes probing a $\mathbb{C}^2/\mathbb{Z}_{N-2}$ singularity. The compactification is done with a $\mathbb{Z}_2$ twist in a discrete symmetry of the 6d SCFT. Particularly,

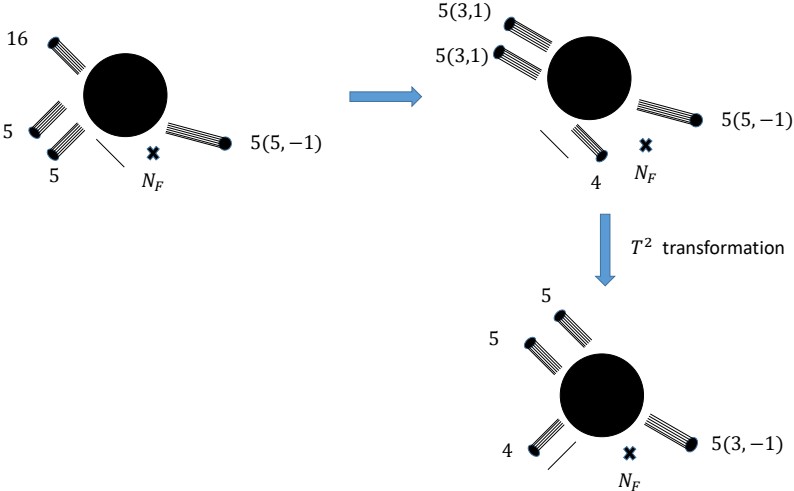

Figure 12: Brane web manipulation for $SU(6)_{\frac{14-N_\mathbf{F}}{2}} + 1\mathbf{AS} + N_\mathbf{F}\mathbf{F}$. The picture on the top left is just the web on the bottom left of Figure 11 for the case of $n = 3$. Moving the $(1, -1)$ 7-brane from top to bottom, we obtain the web on the top right. Performing an $SL(2, \mathbb{Z})$ transformation $T^2$, we end up with the web on the bottom right—this web is a known description of $Sp(5) + 1\mathbf{AS} + N_\mathbf{F}\mathbf{F}$, suggesting that these two theories are dual up to free hypermultiplets.

the 6d SCFT has a 1-dimensional tensor branch along which the theory has a low-energy description as an $SU(N-2) + (2N-4)\mathbf{F}$ gauge theory. In F-theory it is described by:

$$\overset{\mathfrak{su}_{N-2}}{2} \quad [SU(2N-4)].$$

The discrete symmetry in question acts on the gauge theory as charge conjugation. It can also be thought of as the orbifold generalization of the outer automorphism twist compactifications of the $A$ type $(2, 0)$ theory.

The global symmetry of the 6d theory is consistent with the $A^{(2)}_{2N-5}$ found for $SO(N)+(N-2)\mathbf{F}$ using 1-instanton analysis. Additionally, the Coulomb branch dimension agrees between the two, after accounting for the fact that the twist projects out the odd dimension Coulomb branch generators that would have come from the $SU(N-2)$ vector upon reduction to 5d.

## $SU(N)_0 + 1\mathbf{Sym} + (N-2)\mathbf{F}$

This case was also covered in [22]. The 6d lift is the SCFT living on 3 M5-branes probing a $\mathbb{C}^2/\mathbb{Z}_{N-1}$ singularity, which has the F-theoretic description:

$$[SU(N-1)] \quad \overset{\mathfrak{su}_{N-1}}{2} \quad \overset{\mathfrak{su}_{N-1}}{2} \quad [SU(N-1)].$$

The compactification is again done with a twist, generalizing the outer automorphism twist now to the case of the $A_2$ $(2, 0)$ theory, so the twist also acts non-trivially on the tensors. In the low-energy gauge theory description, $(N-1)\mathbf{F} + SU(N-1) \times SU(N-1) + (N-1)\mathbf{F}$, this discrete symmetry is manifested as a combination of quiver reflection and charge conjugation. As consistency checks, we note that the global symmetries and Coulomb branch dimensions agree.

## $SU(N)_0 + 1Sym + 1AS$

We can also consider the case of 5d $SU(N)_0 + 1\mathbf{Sym} + 1\mathbf{AS}$. This theory also lifts to 6d, and behaves differently depending on whether $N$ is even or odd. For odd $N$ we can find the 6d SCFT by generalizing the results of [22]. We find the 6d SCFT to be the one living on $N$ M5-branes probing a $\mathbb{C}^2/\mathbb{Z}_2$ singularity, where again the compactification is done with a $\mathbb{Z}_2$ twist generalizing the outer automorphism twist for the $A_{N-1}$ (2,0) theory. The 6d SCFT has a low-energy gauge theory description as $2\mathbf{F} + SU(2) \times SU(2) \cdots SU(2) \times SU(2) + 2\mathbf{F}$, where there are $N-1$ $SU(2)$ groups. This theory also has the following F-theory description:

$$[SU(2)] \quad \overset{\mathfrak{su}_2}{\mathbf{2}} \quad \overset{\mathfrak{su}_2}{\mathbf{2}} \quad \cdots \quad \overset{\mathfrak{su}_2}{\mathbf{2}} \quad \overset{\mathfrak{su}_2}{\mathbf{2}} \quad [SU(2)],$$

where there are $N-1$ $\mathbf{2}$ curves in the above diagram. The $\mathbb{Z}_2$ discrete symmetry by which we twist acts on the gauge theory as quiver reflection. The web for the even $N$ case has a different form, and appears to go to a different type of theory.

We can again perform several consistency checks for the odd $N$ case. As indicated in Table 1a, the symmetry of the 5d theory is at least $A_1^{(1)} \times U(1)$. This is consistent with the global symmetry of the twisted theory. We also note that the Coulomb branch dimension agrees.

It is instructive to also consider Higgs branch flows. For instance, the 5d theory has a flow from $SU(N)_0 + 1\mathbf{Sym} + 1\mathbf{AS}$ to $SO(N) + 1\mathbf{AS}$ triggered by giving a vev to the symmetric matter, and one to $Sp(\frac{N-1}{2}) + 1\mathbf{S}$ triggered by giving a vev to the antisymmetric matter—these are just the maximally supersymmetric $SO$ and $Sp$ gauge theories. These flows should then also exist in the 6d theory. Indeed we can go on the Higgs branch of the 6d SCFT and flow to either the $A_{N-1}$ or $A_{N-2}$ (2,0) theories. Compactification with a twist of the latter yields maximally supersymmetric $SO(N)$ while the same for the former yields maximally supersymmetric $Sp(\frac{N-1}{2})$ (with theta angle $\theta = \pi$) [29].

This leaves the case of even $N$, which seems to be of a different nature than the theories discussed so far. This is also apparent from the Higgs branch discussion. We again can flow to either the maximally supersymmetric $SO(N)$ or $Sp(\frac{N}{2})$ gauge theory. However the former when $N$ is even lifts to the $D_{\frac{N}{2}}$ type (2,0) theory with untwisted compactification. The latter, alternatively, lifts to twisted compactifications of either the $D_{\frac{N}{2}+1}$ type (2,0) theory or the $A_N$ type (2,0) theory depending on the theta angle of the 5d theory [29]. It seems quite non-trivial to find a theory with this behavior, and therefore we reserve a determination of this case for future work.

## $SU(N)_{\pm\frac{N}{2}} + 1Sym$

To our knowledge this class of theories has not been discussed in the literature. We can argue that this class should exist, at least as a 6d lift gauge theory, as follows. We can try to engineer this theory by using brane webs in the presence of an O7$^+$-plane. The existence of the theory is entirely determined by the asymptotic behavior of the 5-branes. Consider changing the O7$^+$-plane to an O7$^-$-plane with 8 D7-branes. This does not change the monodromy of the object and so does not change the asymptotic behavior of the 5-branes. However, this changes the gauge theory to the previously discussed $SU(N)_{\pm\frac{N}{2}} + 1\mathbf{AS} + 8\mathbf{F}$, which we have argued has a 6d lift. Therefore, it is reasonable to assume that $SU(N)_{\pm\frac{N}{2}} + 1\mathbf{Sym}$ also has a 6d lift. We note that the equivalence in monodromy between an O7$^+$-plane and an O7$^-$-plane with 8 D7-branes is the brane web manifestation of the invariance of the prepotential under exchanging a single symmetric hypermultiplet with a single antisymmetric and 8 fundamental hypermultiplets.

We can then ask what theory is the 6d lift. By examining the brane web in the case $N=3$, we can reformulate the problem in a manner where we are in a position to apply the results

of [22]. We then find that the 6d theory is the $A_4$ $(2, 0)$ theory compactified with an outer automorphism twist. This implies that this theory is dual to maximally supersymmetric $Sp(2)_\pi$, and should thus have an enhancement of SUSY in the UV. Similarly, for $N > 3$, we suspect that the 6d theory is dual to maximally supersymmetric $Sp(N-1)_\pi$ and goes to the $A_{2N-2}$ $(2, 0)$ theory compactified with an outer automorphism twist.

# 6 Discussion and Future Directions

In this paper, we have given a clear physical explanation for why the naive Coulomb branch (i.e. the fundamental Weyl chamber) is not necessarily physical, and why it is therefore necessary to restrict a weakly-coupled gauge theory description to a subset of the Coulomb branch satisfying certain necessary physical criteria. It is necessary that the BPS spectrum consists of degrees of freedom with positive masses and tensions, and the metric be positive definite on this physical region of the Coulomb branch. By extending these ideas, we have proposed a set of conjectures in Section 3.1, 3.2 which we can use for the classification of 5d $\mathcal{N} = 1$ gauge theories with interacting fixed points.

By using these conjectures, we have given an exhaustive classification of 5d $\mathcal{N} = 1$ gauge theories with simple gauge group, which to our knowledge includes all known examples in the literature as a subset—for example, the theories described in [13–17, 21, 22, 26]. Our classification consists of two types of theories, namely standard theories (see Table 1) and exceptional theories (see Tables 3-11). Our strategy has been to identify extremal theories saturating the matter bounds imposed by our physical criteria and then use the fact that all descendants of such extremal theories will have non-trivial UV fixed points.

In practice, we used our main conjecture to test a number of low rank cases explicitly using a symbolic computing tool. Furthermore, we combined Conjectures 2 and 3 to circumvent the problem that a complete analysis of the naive Coulomb branch rapidly becomes intractable as the rank of the gauge group increases; the combination of our conjectures enabled us to make a full classification using a mixture of symbolic and numerical searches. The standard theories are particularly tractable and thus we were able to perform the classification by hand. Computational expense has only hindered our understanding of a few isolated cases of possible exotic theories, as discussed in Section 4.4.

There are a plethora of open questions that remain unresolved. The first question concerns our main conjecture, which asserts that the positivity of monopole string tensions can correctly identify the physical Coulomb branch when all of the mass parameters are switched off. As we explained in Section 3.1, the criterion of positive string tensions can in general miss flop transitions associated to the emergence of massless instantons without tensionless strings (note that the perturbative flop transitions are already captured by the gauge theory interpretation of the prepotential.) However, we remark that the massless instanton spectrum can be captured by an explicit computation of the BPS partition function as described in [37–42], and thus it would be interesting to see if we could test our main conjecture against an explicit computation of the instanton spectrum.

Another puzzle is why, as asserted by Conjecture 2, the positivity of the monopole string tensions guarantees the positivity of the metric. It is unclear from a mathematical perspective why this property should hold true for any gauge theoretic prepotential evaluated on the naive Coulomb branch, and moreover what properties of this function are essential for this phenomenon. Futhermore, Conjectures 1 and 3 seem to lack both mathematical proofs and physical explanations, given that these conjectures involve testing regions of the naive Coulomb branch that are obviously unphysical.

As explained in Section 4.4, there are four potentially exotic theories that we were not able

to confirm due to computational limitations. With a better algorithm or more computational resources, it is possible to check explicitly whether or not these theories are indeed exotic. However, there is a more striking puzzle presented by the notion of exotic theories. From geometric considerations, we expect that truly exotic theories (such as $SU(5)_{\frac{3}{2}}+3\mathbf{AS}+2\mathbf{F}$) undergo some number of flop transitions before reaching the irrational boundaries of the putative physical Coulomb branch. If these theories could be constructed using geometry, this notion could tested explicitly. Alternatively, dual descriptions (as in the case of $5\mathbf{F}+SU(2)\times SU(2)+2\mathbf{F}$, which has a dual description as $SU(3)_{\frac{3}{2}}+9\mathbf{F}$) that are valid in the vicinity of the irrational components of the boundary could provide a clear indication that such theories are well-defined and have an interacting fixed point.

Lingering puzzles aside, there are also many future directions to be explored. For instance, many of the 5d theories appearing in our classification are still lacking a realization either in terms of brane configurations or local Calabi-Yau singularities, and it would be desirable to work out such descriptions in order to provide further evidence for the existence of these theories, as well as to capture the subtleties of the non-perturbative physics. It would be worthwhile to explore the possibility that all of the marginal theories we have identified can be obtained by compactifying some 6d SCFT; developing an understanding of the 6d compactifications, including a possible classification of the intermediary Kaluza-Klein theories, is critical for this purpose.

It would also be quite interesting to further develop a precise geometric understanding of the vanishing monopole string tensions and instanton masses that signal the breakdown of an effective gauge theory description. The field theoretic expression for $\mathcal{F}$ clearly encodes the appearance of massless electric particles in a collection of hyperplanes $w\cdot\phi+m_f=0$ subdividing the fundamental Weyl chamber, across which the Chern-Simons levels jump discontinuously. Geometrically, these "jumps" are interpreted as the result of flops, which can be associated with the interchange of two nonisomorphic resolutions of the same singular Calabi-Yau geometry sharing a common blowdown. Both field theoretically and geometrically these transitions are rather transparent. However it is much more difficult to analyze the extremal transitions associated with the vanishing of monopole tensions or instanton masses away from the conformal point. This point is especially relevant for the distinction between rational and 'irrational' classes characterizing the exotic theories in our classification.

Finally, we would like to extend our classification of simple gauge groups to a full classification of 5d $\mathcal{N}=1$ SCFTs, including quiver gauge theories and non-Lagrangian theories. It is obvious that a physically sensible theory should have a positive definite metric on its physical Coulomb branch having only degrees of freedom with positive messes and tensions. In principle, we can classify general theories with gauge theory description by applying the same techniques we used for the present classification though it may be practically a tremendous task at the moment. A classification of non-Lagrangian theories in the same spirit as our current classification will require a better understanding of how geometric constraints translate into constraints on field theory, much as the case of the geometric description $\mathbb{P}^2$ in $CY_3$ translates to the (formal) field theory description $SU(2)$ with $N_{\mathbf{F}}=-1$.

# Acknowledgements

We would like to thank Thomas T. Dumitrescu, Hirotaka Hayashi, Sheldon Katz, Seok Kim, Sung-Soo Kim, Kimyeong Lee, Kantaro Ohmori, Yuji Tachikawa, Masato Taki, Itamar Yaakov, Futoshi Yagi and Dan Xie for useful comments and discussions. The research of P.J. and H.K. and C.V. is supported in part by NSF grant PHY-1067976. H.K. would like to thank the Aspen Center for Physics Winter Conference on Superconformal Field Theories in $d\geq4$, which is sup-

ported by NSF grant PHY-1066293, and also the Erwin Schrödinger Institute for its workshop on Elliptic hypergeometric functions in Combinatorics, Integrable systems and Physics for hospitality during the final stages of this work. G.Z. is supported by World Premier International Research Center Initiative (WPI), MEXT, Japan.

# A    Mathematical Conventions

Much of the notation and content of the following section is adapted from [43].

## A.1    Root and coroot spaces

Let $R \subset V_R$ be a set of roots in the root space $V_R$, and let $V_R^\vee$ be the coroot space dual to $V_R$. A subset $S \subset R$ is called a system of *simple roots* if all elements $e \in S$ are linearly independent and every root $e' \in R$ belongs to the nonnegative or nonpositive integer linear span of $S$. A *positive root* is a positive linear combination of elements of $S$, and similarly for *negative roots*. As suggested by this definition, there exists a partition $R = R_+ \cup R_- \cup R_0$, where $R_+$ is the set of positive roots, $R_- = -R_+$ and $R_0$ is the set of zero roots. To every choice of $S$, one can associate a *distinguished Weyl chamber* $\mathcal{C}(S)$, where

$$\mathcal{C}(S) = \{\phi \in V_R^\vee \mid \langle \phi, e \rangle \geq 0 \ , \ e \in S\}. \tag{A.1}$$

We define *simple coroots* by

$$\alpha_i^\vee = 2\frac{\alpha_i}{\langle \alpha_i, \alpha_i \rangle}, \tag{A.2}$$

where above $\langle,\rangle$ is the Euclidean inner product. The *Cartan matrix* is defined as:

$$A_{ij} = \langle \alpha_i, \alpha_j^\vee \rangle = 2\frac{\langle \alpha_i, \alpha_j \rangle}{\langle \alpha_j, \alpha_j \rangle} \equiv D_j^{-1}\langle \alpha_i, \alpha_j \rangle. \tag{A.3}$$

In practice, we expand the real vector multiplet scalar vev $\phi$ in a basis of simple coroots:

$$\phi = \sum_i \phi_i \alpha_i^\vee. \tag{A.4}$$

For some computations, we will find it convenient to expand the simple roots in the *Dynkin basis*,[7] or the basis of fundamental weights, as fundamental weights are canonically dual to simple coroots. To illustrate this point, consider the inner product

$$\langle \phi, \alpha_i \rangle = \sum_{j,k} \phi_j A_{ik} \langle \alpha_j^\vee, w_k \rangle = \sum_{j,k} \phi_j A_{ik} \delta_{jk} = \sum_k A_{ik} \phi_k. \tag{A.5}$$

We can apply the same logic above to the scalar product as applied to two coroots:

$$\langle \phi, \phi \rangle = \sum_{i,j} \phi_i \phi_j \langle \alpha_i^\vee, \alpha_j^\vee \rangle = \sum_{i,j} \phi_j D_j^{-1} A_{ji} \phi_i \equiv \sum_{i,j} \phi_j h_{ji} \phi_i, \tag{A.6}$$

where in the above equation we have defined the metric tensor

$$(h^{-1})_{ij} = (A^{-1})_{ij} D_j. \tag{A.7}$$

Therefore, the vector multiplet scalars for the classical kinetic term can be said to be in the (canonical dual of the) Dynkin basis when the quadratic form is given by the inverse of the Lie algebra metric tensor (A.7).

---

[7] Note that the rows of the Cartan matrix $A$ are the simple roots in the Dynkin basis.

## A.2 Changing from Dynkin to orthogonal basis

In some cases, we find it more convenient to work in the orthogonal basis. Therefore, for reference we derive the linear transformation necessary to express the coroot $\phi$ in the orthogonal basis, given an expansion of $\phi$ in terms of simple coroots $\alpha_i^\vee$.

To begin, recall that the Dynkin basis is canonically dual to the basis of simple coroots in the following sense

$$\langle \alpha_i^\vee, w_j \rangle = \delta_{ij}, \tag{A.8}$$

In matrix form, the above equation reads

$$A\Omega^t = I, \tag{A.9}$$

where we define $\Omega$ through the following relation:

$$\Omega = (AA^t)^{-1}A. \tag{A.10}$$

Note that the rows of the above matrix $\Omega$ are the fundamental weights, where $\Omega$ can be viewed as a change of basis matrix from the orthogonal basis to the Dynkin basis:

$$w_i = \sum_j \Omega_{ij}\hat{e}_j. \tag{A.11}$$

The matrix $\Omega$ will be useful in the following derivation.

In order to determine the correct change of basis for $\phi$, we will first convert the simple roots $\alpha_i$ to the orthogonal basis, using the Dynkin basis as an intermediate step. Expanding $\alpha_i$, we find

$$\alpha_i = \sum_j A_{ij}w_j = \sum_{j,k} A_{ij}\Omega_{jk}e_k, \tag{A.12}$$

where $\hat{e}_k$ is a standard basis vector. By inverting $\Omega$ and exploiting the canonical pairing between simple coroots and fundamental weights, we notice that we may write:

$$\langle \phi, \hat{e}_k \rangle = \sum_{i,l} (\Omega^{-1})_{kl} \left\langle \phi_i \alpha_i^\vee, w_l \right\rangle = \sum_i (\Omega^{-1})_{ki}\phi_i, \tag{A.13}$$

where $\Omega^{-1}$ is the pseudo-inverse of $\Omega$. Therefore, the inner product between $\phi$ and a simple root must satisfy the following equation:

$$\langle \phi, \alpha_i \rangle = \sum_k \left( \sum_j A_{ij}\Omega_{jk} \right) \left( \sum_l (\Omega^{-1})_{kl}\phi_l \right) = \sum_j A_{ij}\phi_j. \tag{A.14}$$

In the above equation, we identify

$$\widetilde{A}_{ik} = \sum_j A_{ij}\Omega_{jk}, \tag{A.15}$$

as the components of the matrix whose rows are the simple roots in the orthogonal basis. Therefore, it must be the case that the coroot is expressed in an orthogonal basis as

$$\phi_i = \sum_j \Omega_{ij}\widetilde{\phi}_j. \tag{A.16}$$

In practice, we adopt the notation

$$a_i \equiv \widetilde{\phi}_i, \tag{A.17}$$

to be consistent with commonly used notation in the literature.

### A.3 Lengths of weights, index of a representation

In this section, we expand roots and weights in a basis of simple roots $\alpha_{i=1,\dots,r_G}$ (this basis is referred to as the $\alpha$-basis in [43]). The *length*[8] $l(e)$ of a root $e = \sum_i e_i \alpha_i$ is defined as follows:

$$l(e) = \sum_i e_i \,. \tag{A.18}$$

By linearity, we can extend this notion to weights $w = \sum_i w_i \alpha_i$:

$$l(w) = \sum_i w_i \,. \tag{A.19}$$

The lengths of the weights of an irreducible representation $\mathbf{R}(w^+)$ with highest weight $w^+$ are related to the *quadratic Dynkin index* $c^{(2)}_{\mathbf{R}(w^+)}$ (also referred to as simply 'the index' of the representation $\mathbf{R}(w^+)$) [44]:

$$c^{(2)}_{\mathbf{R}(w^+)} = \frac{\dim(\mathbf{R}(w^+))}{\dim(G)} \langle w^+, w^+ + 2\delta \rangle = C(G) \sum_{w \in \mathbf{R}(w^+)} l(w)^2 \,, \tag{A.20}$$

where in the above expression

$$\delta = \frac{1}{2} \sum_{e \in R_+} e \,. \tag{A.21}$$

For classical Lie algebras $G$, the normalization constant $C(G)$ is chosen so that the index of the fundamental representation is equal to 1:

$$C(G) = \frac{1}{\sum_{w \in \mathbf{F}} l(w)^2} \,. \tag{A.22}$$

However, a different normalization is chosen for the exceptional Lie algebras.

## B  Bound on Matter Representation Dimension

In our classification, we assume that the dimension of the largest matter representation (or half-dimension, in the case of pseudoreal representations) is no larger than the dimension of the adjoint representation. In this section, we explain why this assumption is implied by Conjecture 3, which asserts that the prepotential must be positive on the entire fundamental Weyl chamber.

There exists a point $\phi^*$ in the fundamental Weyl chamber where the inner products $e \cdot \phi, w \cdot \phi$ are equal to the lengths of the corresponding root or weight. This can be understood as follows. In a basis of simple roots, the inner product $e \cdot \phi$ may be expressed as:

$$e \cdot \phi = \sum_{i,j} e_i \phi_j \langle \alpha_i, \alpha_j^\vee \rangle = \sum_{i,j} e_i \phi_j A_{ij} \,, \tag{B.1}$$

and a similar expression holds for $w \cdot \phi$. The conditions placed on $\phi$ parametrizing the fundamental Weyl chamber $\mathcal{C}(S)$ thus translate to the following inequalities:

$$\sum_j A_{ij} \phi_j \geq 0 \,. \tag{B.2}$$

---

[8]In [43], the 'length' of a root or weight is referred to as the 'height'.

Evidently, it is always possible to find a solution $\phi^*$ to the equation

$$\sum_j A_{ij}\phi_j = 1, \quad \text{for all } i, \tag{B.3}$$

since the only constraint on the coefficients $\sum_j A_{ij}\phi_j$ is that they be non-negative, and a Cartan matrix is by definition nonsingular. At this point $\phi^*$, we find

$$e \cdot \phi^* = \sum_j e_j = l(e), \qquad w \cdot \phi^* = \sum_j w_j = l(w). \tag{B.4}$$

Therefore, at infinite coupling $1/g_0^2 = 0$, the prepotential evaluated at this point $\phi^*$ is proportional to

$$6\mathcal{F}(\phi^*) = \sum_{e \in \text{roots}} |l(e)|^3 - \sum_{w \in \mathbf{R}_f} |l(w)|^3, \tag{B.5}$$

and the classical CS term

$$\sum_{w \in \mathbf{F}} (w \cdot \phi^*)^3 = \sum_{w \in \mathbf{F}} l(w)^3 = 0. \tag{B.6}$$

We claim that the difference of sums (B.5) is negative if

$$c_{\mathbf{adj}}^{(2)} < c_{\mathbf{R}_f}^{(2)}. \tag{B.7}$$

(In the case of pseudoreal representations, we replace $c_{\mathbf{R}_f}^{(2)}$ with $\frac{1}{2}c_{\mathbf{R}_f}^{(2)}$.) To see this, assume that the above inequality holds, and observe that $l(w) \in \frac{1}{2}\mathbb{Z}$ for any weight $w$, and therefore up to an overall constant, the quadratic indices

$$c_{\mathbf{R}_f}^{(2)} = \sum_{w \in \mathbf{R}_f} n_w^2, \qquad c_{\mathbf{adj}}^{(2)} = \sum_{e \in \mathbf{adj}} n_e^2, \tag{B.8}$$

where $n_w \in \mathbb{Z}_{\geq 0}$. We therefore find that the quadratic indices are proportional to sums of squares of positive integers. If the above sums satisfy the inequality (B.7), then it must also be true that the sums of cubes $n_w^3, n_e^3$ satisfy the same inequality. But the sums of the cubes of these positive integers are proportional to the sums of the cubes of the lengths. Therefore, it follows that (B.5) must be negative, which violates the condition imposed by Conjecture 3 at the point $\phi^*$.

One can verify that $\dim(\mathbf{R}_f) > \dim(\mathbf{adj})$ implies that $c_{\mathbf{R}_f}^{(2)} > c_{\mathbf{adj}}^{(2)}$, which justifies the exclusion of matter representations with dimension greater than that of the adjoint from our classification.

## C  Convergence of $S^5$ partition function

In this appendix we collect several formulas for the expression of the $S^5$ partition function evaluated using localization. The main result here is that the convergence of the perturbative contribution to the $S^5$ partition function is equal to the demand that the perturbative prepotential be positive on the entire Coulomb branch. We can use this to give another criterion for the UV completeness of 5d gauge theories, which appears to coincide with the other criteria presented in this article, at least in all examples we have checked. This is summarized in the conjectures in section 3.2, particularly conjecture 3.

## C.1 The 5d partition function

The partition function can be represented by the path integral on $S^5$, which in turn can be evaluated using localization; this partition function receives contributions from a perturbative part and a non-perturbative part, where some aspects regarding the latter are still conjectural [45–52]. The general form is suspected to be:

$$Z^{S^5} = \int Z_{\text{pert}} Z_{\text{inst}}^3 , \tag{C.1}$$

where $Z_{\text{pert}}$ is the perturbative part and $Z_{\text{inst}}$ is the non-perturbative contribution. The latter arises from contact instantons [46,52], which fill the $S^1$ in the Hopf-fibration realization of $S^5$ as an $S^1$ fibered over $\mathbb{CP}^2$. They are localized at three points on $\mathbb{CP}^2$ hence the power in the integral. In general they can be expanded in a power series in $\mathbf{q} = e^{-\frac{16\pi^3 r}{g_{YM}^2}}$ :

$$Z_{\text{inst}} = 1 + O\left(e^{-\frac{16\pi^3 r}{g_{YM}^2}}\right) , \tag{C.2}$$

where $r/g_{\text{YM}}^2$ is the ratio of the radius of $S^5$ to the coupling constant.

Let us first ignore the instanton contribution, and consider only the perturbative part. We shall use the expression for it given in [46,47]. The full expression is then:

$$
\begin{aligned}
Z = \frac{1}{|W|} \int_{\text{Cartan}} d\sigma e^{-\frac{4\pi^3 r}{g_{YM}^2} \text{tr}_{\mathbf{F}} \sigma^2 + \frac{\pi\kappa}{3} \text{tr}_{\mathbf{F}} \sigma^3} \det_{\mathbf{adj}}\left(\sin(i\pi\sigma) e^{\frac{1}{2} f(i\sigma)}\right) \\
\times \prod_I \det_{\mathbf{R}_I}\left((\cos(i\pi\sigma))^{\frac{1}{4}} e^{-\frac{1}{4} f(\frac{1}{2}-i\sigma) - \frac{1}{4} f(\frac{1}{2}+i\sigma)}\right) + \text{instanton contributions} ,
\end{aligned}
\tag{C.3}
$$

where $|W|$ is the size of the Weyl group. Also, $\kappa$ is the Chern-Simons level (if present), and $f$ is a function defined by:

$$f(y) \equiv \frac{i\pi y^3}{3} + y^2 \log(1 - e^{-2\pi i y}) + \frac{iy}{\pi} \text{Li}_2(e^{-2\pi i y}) + \frac{1}{2\pi^2} \text{Li}_3(e^{-2\pi i y}) - \frac{\zeta(3)}{2\pi^2} . \tag{C.4}$$

The integral is done over the Cartan subalgebra and the notation $\text{tr}_{\mathbf{R}}, \det_{\mathbf{R}}$ stands for the trace and determinant in the representation $\mathbf{R}$, where $\mathbf{F}$ denote the fundamental represetnation and $\mathbf{adj}$ denotes the adjoint representation.

Ignoring the instanton contributions, the above integral can be recast in the following form [53]:

$$Z = \frac{1}{|W|} \int_{\text{Cartan}} d\sigma e^{-F(\sigma)} , \tag{C.5}$$

where:

$$F(\sigma) = \frac{4\pi^3 r}{g_{YM}^2} \text{tr}_{\mathbf{F}} \sigma^2 + \frac{\pi k}{3} \text{tr}_{\mathbf{F}} \sigma^3 + \text{tr}_{\mathbf{adj}} F_V(\sigma) + \sum_I \text{tr}_{\mathbf{R}_I} F_H(\sigma) . \tag{C.6}$$

We will be concerned with the conditions needed for the convergence of the integral (C.5), for which we will need the asymptotic expansion of $F(\sigma)$, which is:

$$F_V(\sigma) \approx \frac{\pi}{6} |\sigma|^3 - \pi |\sigma| , \tag{C.7}$$

$$F_H(\sigma) \approx -\frac{\pi}{6} |\sigma|^3 - \frac{\pi}{8} |\sigma| . \tag{C.8}$$

One can see that the leading contributions are of order $\sigma^3$, where the contribution of the gauge multiplets generally makes the integral converge while the ones from matter and Chern-Simons terms make it diverge. Thus, the convergence of the integral constrains the number and representations of matter fields and the Chern-Simons level. Hence, it is tempting to conjecture that convergence of the integral may be related to the existence of a non-trivial fixed point.

In the $\sigma \to \infty$ limit the resulting expression for $F(\sigma)$ can be identified with the gauge theory prepotential [54]. This allows us to recast the condition as follows:

$$\sigma \to \infty \Rightarrow F(\sigma) \to \infty \,, \tag{C.9}$$

where $F$ is now the prepotential.

Thus we see that the convergence of the perturbative part of the partition function is equivalent to the demand that the perturbative prepotential be everywhere positive. We stress here that the above (piecewise) expression $F$ for the prepotential is naive because in general a different prepotential is needed for each effective description of the theory valid in a given subset of the naive Coulomb branch. In the absence of instanton corrections, we expect that $F$ will not have a well-defined global expression valid on all of $\mathcal{C}$. In fact, we have encountered numerous examples in our classification program where the prepotential associated to some gauge theory description is only valid on a particular subset of the naive Coulomb branch. Nevertheless, this criterion (i.e. Conjecture 3) leads to physically sensible results consistent with the predictions of our other conjectures.

An interesting case is when the prepotential vanishes along one direction but is positive along all others. In that case the perturbative part of the partition function may diverges by the linear term when $g_{YM} \to \infty$, but should converge for every other value. These cases are then marginal, and in all examples we have checked correspond to 6d lifting theories.

Thus we see that examining the perturbative part of the partition function, we are naturally led to a criterion for the existence of a UV fixed point for a 5d gauge theories. In all the examples we have checked, this criterion gives identical results as the other criteria introduced in this article.

Before ending we wish to discuss the physical aspects of this criterion as well as possible generalizations. A physical argument for this criterion is that the partition function is a characteristic of the theory and must be a finite number so it seems strange for it to diverge. It is known that in some cases partition functions diverges. For instance the index of some 3d theories, the so called 'bad' by [55], diverges. In that case it is due to monopole operators going below the unitary bound, and is believed to signal the appearance of singlets. When the $S^5$ partition function diverges it is expected that likewise there should be a physical explanation for this.

A natural physical explanation then is that the theory needs a UV completion and thus additional degrees of freedom that could curve the divergence. This is also supported as this divergence is of UV type, that is in the limit of infinite Coulomb vev.

There are however several arguments one can raise against this. One such argument is that we only look at part of the expression since we ignored the instanton corrections. So, first, it is possible that even if the perturbative part converges, the instanton contribution diverges. This just means that this condition is not sufficient, but that might be so even when the full partition function converges. It is still reasonable that this is a necessary condition.

However, it is possible that both the perturbative part and some of the instanton contributions diverges, but their sum converges. This seems extremely unlikely since the perturbative part and instanton contributions have different dependence on the coupling constant, which is a mass parameter in 5d controlling the instanton masses. So even if such a cancellation occurs changing the value of the coupling constant should destroy it and leads to a divergence.

Since the partition function must converge for every value of $g_{\text{YM}}$, it still seems to be a good necessary condition.

Special attention should be given to the marginal case when the coefficient of the $\sigma^3$ term vanishes exactly along some direction. In that case the partition function only diverges when $g_{\text{YM}} \to \infty$ so one cannot rule out that the full partition function also converges for $g_{\text{YM}} \to \infty$ where the divergence is canceled by instanton contributions.

Another issue that can be raised is how well can we trust the result for the $S^5$ partition function particularly in the $g_{\text{YM}} \to \infty$ limit. In other words, how well does the gauge theory partition function captures properties of the underlying SCFT when it exists. It is natural to expect that the 5d SCFT should have a physical object that can be identified as its $S^5$ partition functions. This will be a function of various mass deformation, identified with flavor masses and coupling constants in low-energy gauge theory descriptions. The masses in turn can be defined as central charges in the SUSY algebra. Physically we expect this object to be finite for any value of the masses. If we identify this object with the gauge theory partition function then a physical argument for its convergence naturally arises. In that lights, the agreement of this criterion with the others seem to imply that the localized partition function of the gauge theory captures some aspects of the partition function of the underlying SCFT.

Finally we comment on possible generalizations of this criterion. The most obvious one is to generalize to the complete partition function by including the non-perturbative part. As we explained when we discussed the exotic cases, it is expected that there will be some non-perturbative corrections to the perturbative prepotential. Given the identity of the large value behavior of the exponent of the integrand with the perturbative prepotential, and the existence of the non-perturbative instanton corrections, it is tempting to suspect that the latter may give as a window into non-perturbative corrections to the perturbative prepotential. We also only considered the round sphere, while there are generalizations of all the expressions also to the squashed sphere. It might be interesting to explore the generalization to the squashed sphere. It may also be interesting to better understand the relation between the gauge theory partition function and that of the underlying SCFT. We reserve further study of all these issues to future work.

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
