# Peer review of "Towards Classification of 5d SCFTs: Single Gauge Node"

_SciPost Physics, doi:SciPost Phys. 14, 122 (2023)_

## Round 1 · Referee Report · Anonymous (Referee 2) · 2023-2-4

Report

The paper propose a set of criteria necessary for a given 5d gauge theory to be the low-energy effective description of a 5d superconformal field theory (SCFT). This criteria are partially inspired by the string and M-theory constructions of 5d SCFTs. The authors focus on the Coulomb branch effective field theory of non-Abelian gauge theories. Before introducing the criteria, they define the physical Coulomb branch, given by the locus where the masses of the BPS particles and the tensions of the monopole strings are positive.

The first criterion is that the metric has to be positive definite only in the physical Coulomb branch, which is less restrictive wrt the one introduced by Intriligator-Morrison-Seiberg. This clarifies some aspects, which were previously unclear. That is, quiver gauge theories can be low-energy description of 5d SCFTs.
The second main criterion is the convergence of the partition function on the 5-sphere. The convergence of the perturbative part is equivalent to demanding that the prepotential is positive in the physical Coulomb branch. For for large rank gauge theories checking the positivity of the prepotential is easier than checking the convergence of the partition function, thus simplifying the classification algorithm. However, it does not imply the convergence of the non-perturbative part of the partition function.

The authors then classify all possible single node gauge theories (with matter in any representation) that satisfy the criteria above. This gives very useful bounds for the number of hypermultiplets and type of representation for any single node gauge theory.

The authors also propose some conjectures for the existence of the physical coulomb branch. They tested these conjectures in the single gauge node examples previously classified.

The paper is clearly written and contains very useful results. It represent by now a classic reference for the study of 5d SCFTs and the effective gauge theories obtained by relevant mass deformation of the UV SCFT. It has been already widely tested, used and mentioned by other works on this topic. Therefore, the referee recommends this article to be published in SciPost.

---

## Round 1 · Referee Report · Anonymous (Referee 1) · 2023-2-26

Report

The present work extends the field-theoretic criteria by Intrilligator--Morrison--Seiberg for consistent characterization of the Coulomb branches of 5d N=1 SCFTs.
With this extension, the authors can successfully incorporate UV-dualities predicted from 5-brane webs. Furthermore, they demonstrate the efficacy of their proposal by (partially) classifying 5d SCFTs that admit an effective gauge theory description with a simple gauge algebra somewhere on the Coulomb branch.

Being out as a preprint on the arxiv for many years, this work has inspired many follow-up works, in particular in the classification of 5d SCFTs via geometric engineering in M-theory. The results of this paper have since been verified and refined in independent papers. We therefore recommend publication in SciPost.

---

## Editorial Decision

published